# SciNet: Evaluating AI Agents in Relation-Aware Scientific Literature Retrieval

**Chenyang Shao** [1] [2]  **Fengli Xu\*** [1] [2]  **Yong Li** [1] [2]

## Abstract

AI agents have seen widespread adoption in information retrieval for scientific research, giving rise to tools such as Deep Research. However, existing retrieval agents mainly rely on keyword- or embedding-based methods. While effective at capturing content-level similarities, they struggle to understand complex relational networks among scientific papers, such as identifying corroborating or conflicting studies and tracing technological lineages. This fundamental limitation often results in fragmented knowledge structures, misinterpreted research sentiment, and ineffective modeling of collective scientific progress. To address this limitation, we introduce **SciNet**, the first **Sci**entific **Net**work relation-aware dataset for information retrieval agents. Built on a meta-database of 269 million papers across 7 disciplines and containing 8,940 carefully designed tasks, SciNet systematically captures three levels of relational understanding: ego-centric retrieval of papers with novel knowledge structures, pairwise identification of scholarly relationships, and path-wise reconstruction of scientific evolution. Extensive evaluation of three categories of retrieval agents shows that their accuracy on relation-aware tasks often falls below 20%, highlighting a fundamental shortcoming of current retrieval paradigms. Importantly, in a downstream literature review application, agents empowered with SciNet achieve a 25.3% improvement in review quality, highlighting the critical value of relation-aware retrieval for deepening scientific insights. We publicly release SciNet at https://github.com/tsinghua-fib-lab/SciNet to support future research.

[1]Department of Electronic Engineering, BNRist, Tsinghua University, Beijing, China [2]Zhongguancun Academy. Correspondence to: Fengli Xu <fenglixu@tsinghua.edu.cn>.

*Proceedings of the 43rd International Conference on Machine Learning*, Seoul, South Korea. PMLR 306, 2026. Copyright 2026 by the author(s).

## 1. Introduction

The rapid development of AI agents has given rise to advanced research tools, such as *Deep Research* (OpenAI, 2025a), fostering the progress of automated scientific systems, often referred to as AI Scientists (Yamada et al., 2025; Lu et al., 2024). The effective operation of such systems relies on a core capability: high-quality *information retrieval*, which is essential for accurately identifying related work and positioning research projects within the existing literature.

However, information retrieval in scientific research is inherently non-trivial, as it requires understanding not only content-level relevance but also the deeper relational structure of the scientific network. Figure 1 illustrates three representative retrieval cases across different scholarly scenarios: *evaluating knowledge structure*, *understanding peer assessment*, and *capturing collective dynamics*. All these retrieval tasks heavily rely on the scientific network, highlighting the critical importance of relation-aware retrieval.

Despite this necessity, current retrieval agents generally fail to achieve relation-aware retrieval, as illustrated in Figure 1. Embedding-based agents (Beltagy et al., 2019; Cohan et al., 2020; Huang et al., 2020) rely solely on static representations of the literature, limiting retrieval to shallow semantic matching. Meanwhile, Deep Research agents (He et al., 2025; Lála et al., 2023; Skarlinski et al., 2024), despite their iterative pipelines, lack explicit mechanisms to model and fully exploit the relations encoded in the scientific network. Beyond retrieval methods, existing literature retrieval benchmarks similarly overlook deep relational structures in scientific networks. Most benchmarks primarily emphasize semantic precision, evaluating whether retrieved results are topically or domain relevant (Ajith et al., 2024; He et al., 2025). Although STARK introduces a notion of network structure (Wu et al., 2024), it remains limited to structured hops between entities such as authors and affiliations, without capturing the richer relations among publications.

Motivated by these limitations, we propose **SciNet**, the first **Sci**entific **Net**work relation-aware dataset for scientific literature retrieval. SciNet is designed to capture complex relational structures in large-scale scientific networks, enabling systematic analysis of relation-aware retrieval capabilities. SciNet is built upon a comprehensive meta-database of over

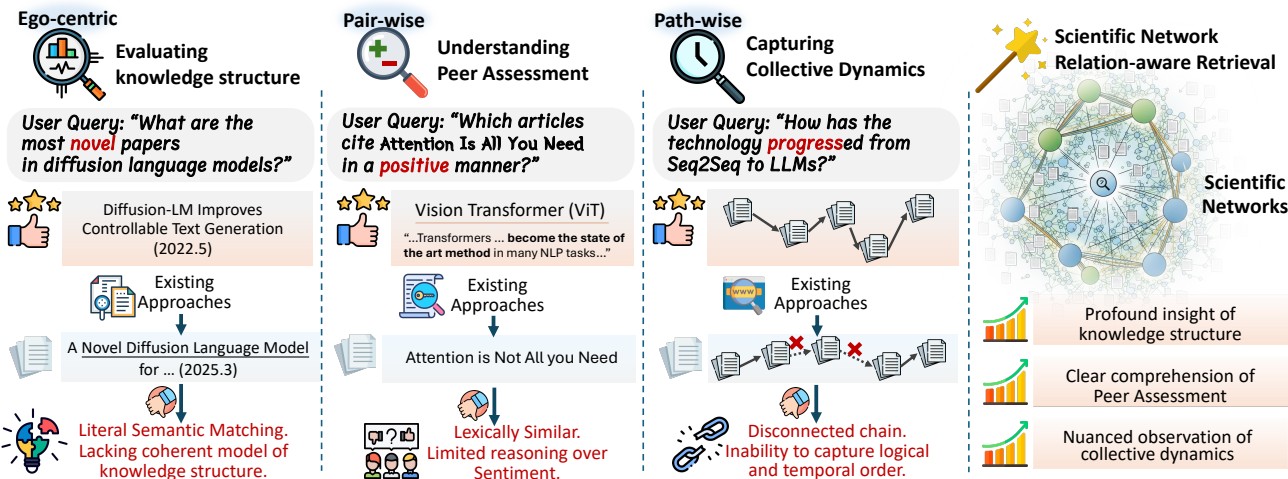

*Figure 1.* Importance of Scientific Networks in Literature Retrieval Scenarios

269 million scientific papers, spanning 7 major scientific domains, including biology, medicine, chemistry, physics, materials science, geology, and artificial intelligence, and further organized into 2,640 fine-grained subfields. From this foundation, we curate a total of 8,940 relation-aware tasks, which can be grouped into three categories according to the level of relational abstraction:

- **Ego-centric**: Retrieving papers based on intrinsic scientific properties, such as identifying the most novel or disruptive work within a research area. We draw on authoritative metrics from the field of scientometrics (Uzzi et al., 2013; Funk & Owen-Smith, 2017) to quantify the novelty and disruptiveness of papers.
- **Pair-wise**: Identifying the specific relational context between two papers, such as whether one supports, contradicts, or extends the other. We perform analysis, identification, and reasoning based on the full text of papers to accurately determine the relationships between them (Ritchie et al., 2008; Hernández-Alvarez & Gomez, 2016; Hsiao & Torvik, 2023).
- **Path-wise**: Retrieving citation paths that reflect the evolution of scientific ideas, such as reconstructing the development trajectory from a foundational concept to a state-of-the-art method. We identify abundant evolutionary pathways through extensive and in-depth exploration within large-scale citation networks (Chen, 2017; Zhang et al., 2017).

Using SciNet, we conduct a systematic evaluation of eight retrieval methods, covering three representative paradigms: (1) embedding-based retrieval, (2) agentic retrieval, and (3) deep research pipelines. Results show that existing approaches consistently underperform across all three categories of tasks. We further demonstrate the practical value of SciNet in a downstream application of *literature review*. Retrieval agents empowered by SciNet generate substantially higher-quality literature summaries, underscoring the

importance of effectively exploiting the literature network. Our contributions can be summarized as follows:

- We systematically define three levels of scientific relations: ego-centric, pair-wise, and path-wise, to characterize how scholarly papers relate to each other.
- We construct **SciNet**, the first large-scale relation-aware dataset for literature retrieval, which provides standardized queries and evaluation protocols to support analysis of relational retrieval capabilities.
- Through extensive evaluation of 8 retrieval methods, we quantify the limitations of current approaches and demonstrate the critical importance of relational retrieval. Additional experiments further validate the benefits of the literature network for downstream applications.

**Conflict of Interest Disclosure.** The authors declare that they have no financial conflicts of interest related to this paper.

## 2. Dataset Overview & Evaluated Models

### 2.1. Dataset Overview

*Scientific network* refers to a semantically enriched graph of scholarly publications, constructed from citation relations among papers and citation contexts within documents, so as to capture not only the presence of citation links but also their underlying semantic nature. To build such a network, we first leverage OpenAlex (RELEASE 2025-07-07), a comprehensive open scholarly dataset, to obtain citation relations among scientific papers. We downloaded the complete data snapshot directly from its official Amazon S3 bucket[1] , which contains metadata for 269,091,010 papers,

---

[1]https://docs.openalex.org/download-all-data/download-to-your-machine

including titles, abstracts, authorship, citations, and publication information.

We focus on seven representative scientific domains, biology, medicine, chemistry, physics, materials science, geology, and artificial intelligence, which together cover the vast majority of the natural sciences. For each domain, we identify and curate finer-grained subfields using OpenAlex's topic classification system in combination with manual review and aggregation (e.g., under AI: three-dimensional reconstruction, tool-augmented reasoning; under biology: Industrial Microbiology, Metabolic Engineering, etc.). The complete list of subfields is provided in our repository. We organize the full set of 269 million papers according to these domains and subfields, resulting in what we believe to be the largest manually validated corpus for literature retrieval to date. Additionally, we incorporate the full arXiv PDF corpus (as of July 7, 2025), as well as all open-access papers available through OpenAlex, to support auxiliary text extraction and validation.

Based on this corpus, we construct **8,940 high-quality queries** spanning three relational tasks: 2,640 (29.5%) for ego-centric retrieval, 4,200 (47%) for pair-wise relation identification, and 2,100 (23.5%) for path-wise evolutionary analysis. Queries are first generated using structured rules leveraging citation and topical information, and subsequently validated through expert manual review to ensure both coverage and reliability. A detailed introduction of our dataset construction pipeline, including automated computation, rule-based derivation, and human quality assurance mechanisms, is provided in Appendix A.5.

### 2.2. Evaluated Models

We evaluate 8 retrieval models across 3 categories:

**Category I: Retrieval via Embedding Models: SciBERT** (Beltagy et al., 2019) is the first embedding model specifically trained on scientific literature. It was pre-trained on a corpus of 3.17 billion tokens, predominantly from the biomedical domain. More recently, with the rapid advances in large language models (LLMs), their embedding layers have also been regarded as reliable embedding models. So we include the newly released and powerful **Qwen3-8B-Embedding** (Zhang et al., 2025) model in the evaluation.

**Category II: Retrieval via Agentic Models:** This category encompasses frameworks that employ agentic workflows for information retrieval and synthesis. We include **paperQA2** (Skarlinski et al., 2024), a recently released system by FutureHouse designed for high-accuracy, retrieval-augmented QA over scientific documents. Its agent-driven framework integrates vector retrieval with LLM-based comprehension, first segmenting the corpus into discrete text chunks and indexing them individually, then executing a pipeline that includes evidence gathering and answer genera-

tion with explicit citation support. Also selected is **PaSa** (He et al., 2025), which operates through a Crawler and a Selector. The Crawler autonomously generates search queries from user input, retrieves relevant papers, and iteratively expands the search scope through citation tracking. The Selector then evaluates the relevance of the retrieved papers to the query. Further included are **gpt-4o-mini-search** and **gpt-4o-search** (OpenAI, 2025b), which leverage LLMs to perform multi-step reasoning. These models are equipped with powerful web search tools, enabling them to autonomously generate queries, retrieve information from diverse online sources, and synthesize responses.

**Category III: Retrieval via Deep Research Agents:** We selected **o4-mini-deep-research** and **o3-deep-research** (OpenAI, 2025a). Deep Research agents operate through a deeply iterative pipeline, in which they autonomously decompose complex queries into sub-tasks and dynamically adapt retrieval strategies. They can execute dozens of iterative search-and-reasoning cycles before the final answer.

Further implementation details, including embedding indexing strategies and model deployment configurations, are provided in Appendix A.2.

## 3. Ego-centric Relation: Evaluating Knowledge Structures via Scientometrics

### 3.1. Evaluation Protocol

**Construction:** Beyond merely finding related papers, deep scientific inquiry often requires evaluating the intrinsic scholarly value of individual publications, such as identifying truly novel or disruptive work. To address this need, we introduce Ego-centric Retrieval, a category focused on assessing papers based on their knowledge structure. Literature attributes like novelty often arise from distinctive configurations in a paper's knowledge structure, such as the pioneering combination of concepts from previously disconnected fields. By leveraging scientometric indicators, we can quantify such structural characteristics to answer queries like, "Which is the most novel paper in diffusion language models?", a task beyond semantic matching, as it requires comprehension of abstract, structure-derived properties.

To perform this task, we make use of the collective knowledge embedded in citation networks. For example, a paper's citation patterns, how it is cited by later work, can serve as a reliable indicator of its novelty, and disruptiveness. Building on this idea, we convert two established scientometrics indicators, the *novelty* and the *disruption index*, into concrete retrieval queries. This allows us to evaluate whether retrieval models can correctly interpret and respond to queries aimed at capturing the intrinsic scientific value of papers.

Specifically, we draw on the method proposed by Uzzi et

| Category | Models | Novelty-SoS | Novelty-LLM | Novelty-Recall@50 | Disruption-SoS | Disruption-LLM | Disruption-Recall@50 |
|---|---|---|---|---|---|---|---|
| **Embedding** | SciBERT | 3.675 | 2.384 | 0.00% | 4.270 | 2.213 | 0.00% |
| | Qwen3-Embed | 4.105 | 2.843 | 0.18% | 3.640 | 3.027 | 2.28% |
| **Agentic** | PaSa | 4.643 | 5.062 | 0.10% | 6.048 | 3.829 | 2.66% |
| | PaperQA | 5.519 | 5.501 | 0.20% | 5.673 | 3.970 | 2.35% |
| | gpt-4o-mini-search | 6.363 | 6.199 | 0.73% | 6.642 | 5.887 | 3.39% |
| | gpt-4o-search | 6.440 | 6.299 | 1.17% | 6.685 | 6.024 | 4.47% |
| **DeepResearch** | o4-mini-deep-research | 6.838 | 6.474 | 1.33% | 6.928 | **7.054** | **4.71%** |
| | o3-deep-research | **6.951** | **6.562** | **1.47%** | **6.960** | 6.982 | 3.94% |

*Table 1.* Performance of Frontier Models and Agents on Ego-Centric Tasks

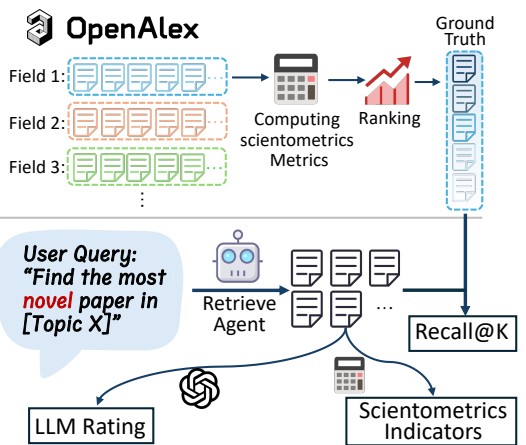

*Figure 2.* Evaluation Protocol of Ego-Centric Retrieval.

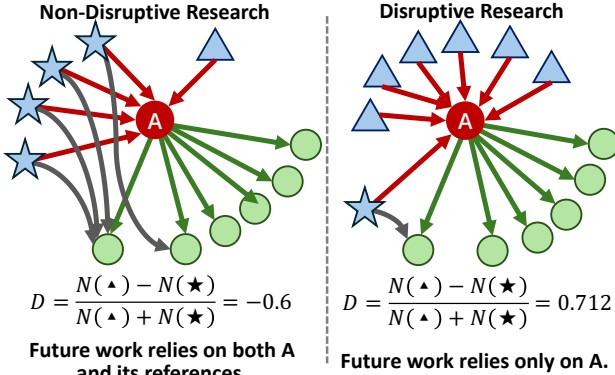

*Figure 3.* Intuitive Illustration of Non-Disruptive and Disruptive Research.

al. (Uzzi et al., 2013) for measuring **novelty**: by analyzing co-citation pairs within a paper's references, they quantify the extent to which the work combines rare or "atypical" knowledge components. This is calculated by converting each co-citation pair's frequency into a Z-score relative to the disciplinary norm, with the paper's final novelty score being the 10th percentile ($p_{10,z}$) of these scores. A lower score thus signifies a more novel combination of knowledge. In parallel, we adopt the **disruption index** introduced by Funk and Owen-Smith (Funk & Owen-Smith, 2017). This metric evaluates whether subsequent publications continue to cite both the focal paper and its predecessors, or instead shift to citing only the focal paper. The index is calculated as $(N_i - N_j)/(N_i + N_j)$, where $N_i$ is the count of papers citing only the focal work and $N_j$ is the count citing both the focal work and its references, thereby characterizing whether the work extends existing trajectories or fundamentally disrupts prior research. An intuitive comparison illustrating the disruption index is provided in Figure 3.

**Evaluation:** For evaluation, each query is used to retrieve a ranked list of 50 candidate papers. We employ a multifaceted assessment strategy covering three distinct aspects. First, for our scientometrics indicators, we calculate the raw novelty and disruption scores for each retrieved pa-

per. The raw novelty score is a Z-score that is theoretically unbounded, where more negative values indicate higher novelty, while the disruption index ranges from -1 (consolidating) to +1 (disruptive). Given the distinct and unintuitive scales of these raw scores, we convert each into a percentile rank against a global reference corpus of millions of papers. The average of these percentile ranks for the candidates constitutes our final ***Novelty-SoS*** and ***Disruption-SoS*** metrics.

Second, we incorporate LLMs (GPT-5) to provide complementary semantic judgments, yielding the ***Novelty-LLM*** and ***Disruption-LLM*** metrics. These models assign a score from 1 to 10 for each concept based solely on the paper's title and abstract. Third, we establish ground truth labels to measure retrieval performance. For each of the subfields, we identify the top 50 most novel and top 50 most disruptive papers based on their scientometric ranks, creating two distinct ground truth sets. Performance is then measured using ***Novelty-Recall@50*** and ***Disruption-Recall@50***, which evaluate the system's ability to include these key papers within its top 50 results.

### 3.2. Experimental Results

As shown in Table 1, a clear performance hierarchy is evident across all evaluation metrics for ego-centric retrieval. Deep Research systems (e.g., o3-deep-research) consis-

tently achieved the best results, followed by web search-based agentic models (e.g., gpt-4o-search), while other models demonstrated substantially weaker performance. This demonstrates that agentic workflows and flexible use of web tools can effectively enhance performance in literature retrieval. Nevertheless, even the top-performing systems struggled significantly according to recall-based evaluation: the best recall@50 for novelty was only 1.47%, and for disruption only 4.71%, indicating that over 95% of truly groundbreaking papers were missed by all systems.

This pattern was consistent across both scientometrics scores and LLM-based assessment, revealing that current retrieval approaches are misaligned with the demands of scientific practice, where accurate assessment of papers based on their intrinsic scientific properties is essential. The observed failures underscore the necessity of developing relation-aware retrieval models capable of understanding scholarly networks and capturing papers' deeper value. To intuitively illustrate these limitations, we provide an in-depth case study on retrieving disruptive papers in Direct Preference Optimization in Appendix B.1.

## 4. Pair-wise Relation: Understanding Peer Assessment through Citation Contexts

### 4.1. Evaluation Protocol

Building upon the scientometrics indicators discussed previously, which primarily focus on the statistical properties of individual nodes within the scholarly network, this section extends the analysis to pairwise relations between papers, with particular emphasis on fine-grained semantic associations derived from citation contexts (peer assessment). Accurately identifying such relational information is critical for multiple downstream scientific applications: it enables high-precision literature recommendation by moving beyond topical similarity to capture nuanced scholarly dialogues; it significantly improves the quality of retrieval-augmented generation (RAG) systems by providing evidence chains with explicit sentiment and contextual labels; and it supports the construction of richly structured knowledge graphs that reflect the true discursive landscape of a field, facilitating advanced analyses such as trend detection, controversy mapping, and knowledge gap identification.

To operationalize this focus, we designed a suite of pairwise retrieval tasks encompassing two critical types of scholarly relationships. The first type involves **sentiment-oriented queries**, such as "Which papers cite Paper XX positively?", requiring systems to distinguish between critical, supportive, or neutral citations based on contextual sentiment. The second type targets **context-based co-mention queries**, exemplified by "Which papers are frequently mentioned together with Paper XX within the same paragraph?". This

task demands the identification of papers jointly referenced within a coherent narrative segment (e.g., a paragraph in the related work section), thereby capturing methodological comparisons, or thematic contrasts within the scientific literature.

**Evaluation:** Retrieval quality is assessed through four complementary metrics. *Cite-Acc* measures whether a retrieved paper is actually cited by the query paper. *Cite-Sentiment* extends the evaluation by analyzing the sentiment of the citation: each retrieved paper's PDF is obtained from arXiv and parsed with *GROBID* (Lopez & Romary, 2013), which provides both the reference list and in-text citation links; verified citations are then traced to their surrounding paragraphs, where GPT-5 determines whether the citation is positive, negative, or neutral. *CoMention-Acc* captures contextual co-citation by checking in the citation network whether a source article cites both the retrieved paper and the query paper. Building on this, *CoMention-Paragraph* requires stronger evidence by parsing the co-citing article with *GROBID* to confirm that both citations not only appear but also co-occur within the same paragraph, thereby ensuring that co-citation evidence is grounded in explicit textual context rather than inferred solely from the network.

**Human Validation of Sentiment Annotation:** To validate the reliability of the LLM sentiment classification, we conducted a human validation study. Experts manually annotated the sentiment of 200 sampled citation contexts. The comparison against GPT-5's annotations demonstrated a 98% agreement rate, with only four minor discrepancies on the boundary of positive and neutral sentiments (three cases where the LLM predicted "neutral" while humans labeled "positive," and one case where the GPT-5 predicted "positive" while humans labeled "neutral"). This strong alignment confirms the high accuracy and reliability of the LLM-generated annotations for citation sentiment.

### 4.2. Experimental Results

Table 2 presents the performance of different systems on the pair-wise tasks. The results indicate that current agentic retrieval approaches, including both web search tools and reasoning-based agents, provide improvements in retrieval quality, with deep research platforms yielding even more substantial gains. For example, *Cite-Acc* reaches around 46% for agentic models and exceeds 60% for deep research systems, while *CoMention-Acc* can be as high as 76%. These findings suggest that leveraging external search capabilities or reasoning mechanisms enables models to more effectively identify citation links and co-citation patterns compared to purely embedding-based methods.

However, substantial challenges remain. Metrics such as *Cite-Sentiment* and *CoMention-Paragraph* continue to show low performance across all approaches, indicating that cap-

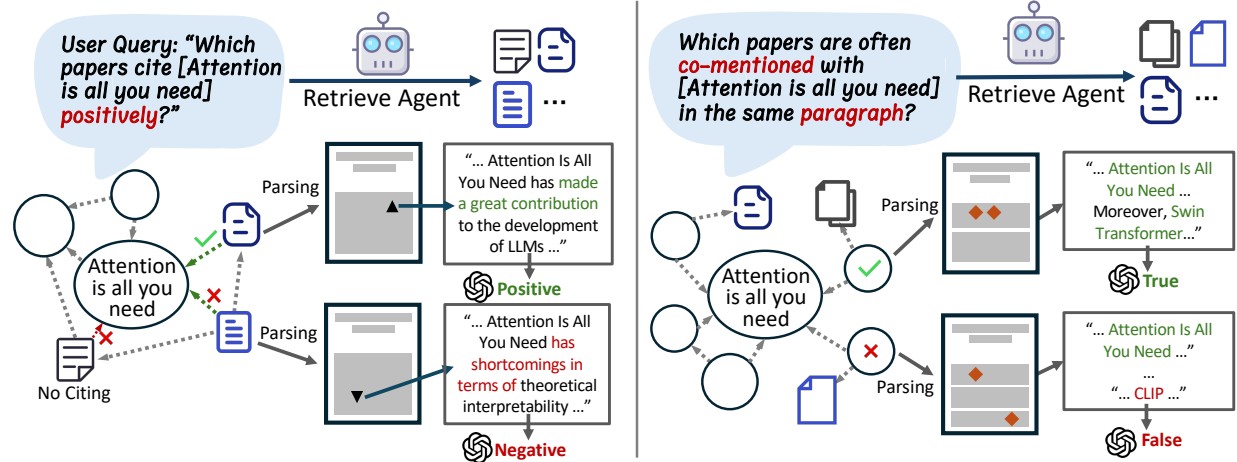

*Figure 4.* Evaluation Protocol of Pair-Wise Retrieval.

| Category | Models | Cite-Acc | Cite-Sentiment | CoMention-Acc | CoMention-Paragraph |
|---|---|---|---|---|---|
| **Embedding** | SciBERT | 2.13% | 0.00% | 0.00% | 0.00% |
| | Qwen3-Embed | 11.34% | 2.89% | 22.93% | 5.23% |
| **Agentic** | PaSa | 19.54% | 5.26% | 24.19% | 7.48% |
| | PaperQA | 8.43% | 3.44% | 8.40% | 7.21% |
| | gpt-4o-mini-search | 46.24% | 13.31% | 66.17% | 6.93% |
| | gpt-4o-search | 48.65% | 10.33% | 69.94% | 7.50% |
| **DeepResearch** | o4-mini-deep-research | 62.93% | **17.83%** | 66.70% | 11.32% |
| | o3-deep-research | **62.73%** | 13.68% | **75.68%** | **17.09%** |

*Table 2.* Performance of Frontier Models and Agents on Pair-Wise Tasks

turing citation sentiment and grounding co-mentioned papers within the same paragraph remain difficult. Overall, while advanced retrieval methods enhance citation detection, relational and context-aware reasoning is still far from solved, clearly highlighting the necessity of leveraging the literature network for document retrieval.

## 5. Path-Wise Relation: Capturing Collective Dynamics of Scientific Evolution

### 5.1. Evaluation Protocol

**Construction:** The previously introduced ego-centric and pair-wise tasks assess models' ability to capture intrinsic properties and binary relations. However, scientific progress typically unfolds as an evolving narrative, where new ideas build upon prior work in multi-step trajectories, forming the collective dynamics of scientific knowledge. To capture this essential aspect, we propose our third category: *Path-Wise Retrieval*, which evaluates whether a system can reconstruct the evolutionary path connecting a sequence of papers. For example, a researcher might ask: "What are the key milestones linking the seminal Transformer paper to today's large language models?" Answering such queries requires understanding not just paper relevance, but also the logical and citational dependencies that form a coherent

developmental chain.

The importance of this task lies in its centrality to literature reviews and research trend analysis. A system that merely outputs unordered related papers cannot reveal the intellectual structure of a field. Yet, existing retrieval methods are almost entirely incapable of constructing such paths, as they lack mechanisms for modeling temporal progression or causal reasoning in scholarly lineage. By introducing the path-wise task, our dataset offers the first rigorous testbed for evaluating retrieval systems on their ability to reconstruct scientific evolution, pushing them beyond shallow retrieval toward genuine knowledge synthesis.

To construct meaningful technological evolution queries, we leveraged the aforementioned subfields. For each subfield, we retrieved the top 50 most-cited papers of all time (treated as classical papers) and the top 10 most-cited papers since 2024 (treated as emerging papers). By randomly pairing classical and emerging papers, we generated a large set of candidate pairs. We then applied the OpenAlex citation network to filter out paper pairs that are topologically connected, followed by manual inspection to ensure that the paired papers remain thematically coherent. This yielded a total of 2,100 high-quality queries, such as: "What is the most influential citation path from *Attention Is All You Need* to *An Image is Worth 16x16 Words: Transformers for Image*

| Category | Models | Consistency | Connectivity | Rationality-LLM | Rationality-Human |
|---|---|---|---|---|---|
| **Embedding** | SciBERT | 0% | 0% | 1.223 | 1.72 |
| | Qwen3-Embed | 2.18% | 3.13% | 3.031 | 2.20 |
| **Agentic** | PaSa | 4.22% | 1.93% | 2.356 | 2.37 |
| | PaperQA | 5.57% | 2.67% | 1.998 | 2.31 |
| | gpt-4o-mini-search | 52.84% | 7.41% | 4.505 | 5.03 |
| | gpt-4o-search | **65.52%** | 10.30% | 4.653 | 5.41 |
| **DeepResearch** | o4-mini-deep-research | 62.36% | 12.08% | 6.705 | 6.74 |
| | o3-deep-research | 63.28% | **14.54%** | **6.893** | **7.06** |

*Table 3.* Performance of Frontier Models and Agents on Path-Wise Tasks

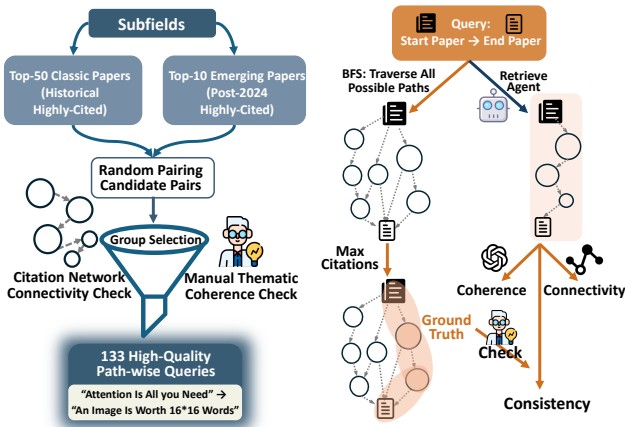

*Figure 5.* Evaluation Protocol of Path-Wise Retrieval.

*Recognition at Scale*?"

**Evaluation:** For each query, we constructed the ground truth path using a breadth-first search (BFS) algorithm. Specifically, we enumerated all connecting paths between the two endpoint papers and computed the cumulative citation count of all papers along each path. The path with the highest total citation count was selected as the candidate trajectory. This approach is well justified, as it closely parallels the notion of collective credit allocation proposed by Hua-Wei Shen et al. (Shen & Barabási, 2014), where citations are interpreted as community votes that represent collective recognition of a research trajectory. Subsequently, experts conducted logical coherence checks on these candidate paths, manually filtering out meaningless trajectories that were topologically connected but scientifically disjointed, to establish the final ground truth.

We evaluated retrieval results along three complementary dimensions. First, **Consistency** measures the degree of overlap between the retrieved path and the ground-truth path. Second, **Connectivity** evaluates whether the retrieved papers form a connected citation path linking the query endpoints within the citation network. Third, **Rationality** measures the plausibility of the retrieved path. We first evaluate this using an LLM, which is prompted with the titles and abstracts of the retrieved papers to judge whether the sequence forms a coherent and reasonable evolutionary narrative. To further enhance reliability, we complement this with a human evaluation: for a subset of 50 representative queries, three AI-expert annotators independently scored the retrieved paths from each model on a 1–10 scale based on rationality. The scores are then averaged to form a *Rationality-Human* metric, which is included as an additional column.

## 5.2. Experimental Results

Results shown in Table 3 reveal a pronounced performance gap between traditional embedding-based methods and more advanced retrieval paradigms on the path-wise task. Embedding models such as SciBERT and Qwen3-Embed essentially fail, achieving near-zero *Consistency* and *Connectivity*, alongside very low scores in both LLM-judged (*Rationality-LLM*) and human-judged (*Rationality-Human*) evaluations. This indicates that these models cannot capture sequential dependencies or reconstruct coherent scientific trajectories beyond surface-level semantic similarity.

In contrast, web search and deep research systems demonstrate substantially stronger performance. Models such as *gpt-4o-search* and *o3-deep-research* achieve over 60% *Consistency*, successfully retrieving papers that lie along true evolutionary paths. However, a significant discrepancy remains between candidate retrieval and logical linking; for instance, while *o3-deep-research* achieves a leading *Consistency* of 63.28%, its *Connectivity* is capped at 14.54%, highlighting that even top-tier models struggle to maintain explicit citational chains. Furthermore, the superiority of DeepResearch models is quantified by the *Rationality-Human* metric, where *o3-deep-research* scores 7.06, nearly quadruple the performance of baseline embedding models, confirming a much higher capacity for generating plausible scientific narratives. These results highlight that reconstructing intellectual lineages demands relation-aware retrieval, and the path-wise task offers a rigorous framework for evaluating advanced retrieval beyond surface-level semantic matching. Additionally, a detailed case study demonstrating the reconstruction of evolutionary trajectories, specifically from the Neocognitron to the Swin Transformer, is provided in Appendix B.2.

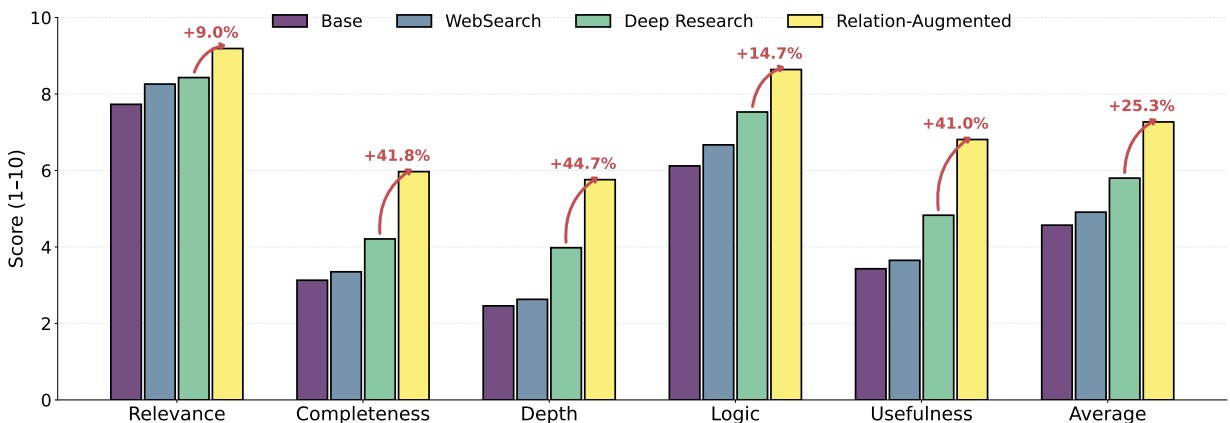

*Figure 6.* LLM Evaluation of Survey Generation Quality.

# 6. Demonstrating the Practical Value of SciNet via Downstream Applications

To demonstrate the practical value of our dataset, we conducted a case study on **automatic literature review**. Through this experiment, we aim to illustrate, from an application perspective, the critical importance of relation-aware retrieval.

**Experimental Setup:** We selected a set of 40 representative queries from our path-wise dataset, each corresponding to a research path with ground-truth papers and abstracts. For each query, the ground-truth sequence provides a reference trajectory of the evolution of ideas, allowing for systematic evaluation of survey generation methods.

Four different approaches were evaluated: *Base LLM*: An LLM that generates surveys solely from the input query and paper abstracts, without any external retrieval or ground-truth information. *Search-enabled LLM*: An LLM leverages web-based literature retrieval tools to generate survey reports without access to ground-truth paper sequences. *Deep Research System*: Our proposed system, which explicitly models literature evolution and performs multi-hop retrieval over relational structures. *Base LLM with Ground Truth*: The same model as above, but provided with the ground-truth paper sequences for each query to assess the upper-bound performance achievable when the full evolution path is known. All methods used identical prompt templates, emphasizing structured survey writing in academic Markdown style, highlighting the progression and connections between papers, and focusing on relevance, completeness, depth, logical flow, and overall usefulness.

**Evaluation Metrics:** To quantify survey quality, we adopted two complementary protocols: (1) *LLM Automatic scoring*: Each generated report was evaluated along five dimensions: *Relevance*, *Completeness*, *Depth*, *Logical Consistency*, and *Usefulness*. Scores were assigned on a scale of 1–10, and aggregated averages across queries were com-

| Method | Avg. Rank ↓ | #Rank=1 | #Rank=2 | #Rank=3 | #Rank=4 |
|---|---|---|---|---|---|
| Ground Truth | **1.33** | 30 | 7 | 3 | 0 |
| Deep Research System | 2.20 | 6 | 22 | 10 | 2 |
| Base LLM | 3.00 | 3 | 6 | 19 | 12 |
| Search-enabled LLM | 3.58 | 1 | 3 | 8 | 28 |

*Table 4.* Human preference rankings across 40 queries. Lower average rank indicates better overall preference.

puted for each method (Figure 6). (2) *Human preference ranking*: Three domain experts were asked to comparatively rank the four systems' outputs for each query (from most to least useful). Table 4 summarizes the distribution of ranks and average scores.

**Results and Analysis:** Across both automatic evaluation metrics (Figure 6) and human preference rankings (Table 4), the relation ground-truth augmented model attains the highest scores on every evaluated dimension, confirming the clear upper bound when accurate research trajectories are available. The DeepResearch system is the strongest baseline: it substantially outperforms the rest of the models, most notably in Completeness and Depth. By contrast, the Base LLM and Search-enabled LLM lag behind; the search-enabled model attains comparatively high relevance but exhibits low completeness and depth, while the base model shows only modest gains in logical coherence. The concordance between automatic evaluation and human rankings indicates that relation-aware retrieval materially improves survey quality, yet the remaining gap in completeness and depth between Deep Research and Ground Truth highlights the need for better modeling of literature relations.

Taken together, these findings underscore that **capturing literature relations is critical for practical downstream applications**. Systems with access to richer relational context produce more coherent, informative, and useful survey reports. Beyond literature review, literature relations can also provide significant benefits in other applications such as automated experiment design, innovation ideation, and scientific knowledge discovery. This highlights the practical

value of our dataset in supporting the development of systems capable of nuanced scientific reasoning that directly benefits real-world applications. Finally, we provide a detailed analysis of deployment costs and retrieval latency for scientific networks in Appendix A.3.

## 7. Related Works

Several benchmarks have been proposed to advance scientific literature retrieval. Here, we review representative efforts and highlight how our work emphasizes relational understanding beyond thematic or entity-centric retrieval.

LitSearch (Ajith et al., 2024) constructs queries using two complementary strategies. *Inline-citation questions* sample paragraphs with citations from research papers, then GPT-4 rewrites them into literature search questions answerable by the cited works. *Author-written questions* are crafted by paper authors, guided by realism, specificity, and resistance to trivial keyword-based resolution.

The PASA (He et al., 2025) benchmark similarly generates queries from *related work* sections using LLMs (e.g., GPT-4o) and expands candidate sets via conventional and academic search engines, search-augmented ChatGPT, and LLM rewriting, with manual expert filtering. Both PASA and LitSearch primarily focus on topical localization, identifying papers in specific domains. In contrast, our dataset emphasizes reasoning over scholarly relationships, such as methodological influence, disruptive contributions, and conceptual development. STARK (Wu et al., 2024) builds a semi-structured database from the Microsoft Academic Graph, supporting structured knowledge queries that require multi-hop reasoning over predefined entities. Unlike STARK, our dataset evaluates the discovery of implicit, semantically rich connections among scientific works, providing a more natural testbed for deep scientific reasoning.

For a more comprehensive comparison between SciNet and existing scientific retrieval benchmarks across corpus scale, task focus, and supervision signals, please refer to Appendix A.1.

## 8. Conclusion

In this paper, we propose a dataset, **SciNet**, for relation-aware retrieval in scientific literature. SciNet evaluates retrieval systems across three granularities: **ego-centric** tasks that focus on individual papers' intrinsic scientific properties, **pair-wise** tasks that assess the relationships between two papers, and **path-wise** tasks that reconstruct citation paths to capture the evolution of scientific ideas. Our experiments demonstrate that current retrieval methods struggle to capture these relational structures, and that this deficiency can substantially degrade downstream applications such as

literature review. By emphasizing relational understanding over isolated texts or semantic similarity alone, SciNet highlights the practical necessity of integrating scholarly relations, providing a foundation for developing retrieval systems capable of nuanced scientific reasoning and more reliable knowledge synthesis.

## Impact Statement

This paper presents work whose goal is to advance the field of machine learning. There are many potential societal consequences of our work, none of which we feel must be specifically highlighted here.

## Acknowledgements

This work was supported in part by the National Natural Science Foundation of China (Grant No. 62472241), in part by the Tsinghua-Toyota Joint Research Center, and in part by the Zhongguancun Academy (Grant No. C20250401).

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

# A. Supplementary Details and Discussion

## A.1. Extended Comparison with Existing Datasets

To further clarify the contributions of SciNet, we provide a comprehensive comparison with existing mainstream scientific retrieval and RAG benchmarks, including LitSearch (Ajith et al., 2024), PASA (He et al., 2025), and STARK (Wu et al., 2024). As summarized in Table 5, our benchmark significantly differs from prior work across three key dimensions:

**Massive Corpus Scale.**    While existing benchmarks often focus on a specific subset of papers (e.g., PASA focuses on several high-tier AI conferences) or limited graph nodes (e.g., STARK contains approximately 3M nodes), SciNet indexes the full OpenAlex corpus of **269,091,010 papers**. This unprecedented scale fully simulates real-world academic discovery environments, providing the most authentic assessment of an agent's capabilities in practice.

**From Semantic Matching to Relation-Aware Reasoning.**    Most existing benchmarks focus on semantic retrieval (finding relevant text) or information extraction (identifying specific entities). In contrast, SciNet introduces *Relation-aware Scientific Reasoning*. Our tasks (Ego-centric, Pair-wise, and Path-wise) require models to understand the structural topology of knowledge, i.e., identifying paradigm-shifting papers or reconstructing technological lineages, which cannot be solved by simple keyword matching or semantic similarity.

**Objective and Robust Supervision Signals.**    Unlike benchmarks that rely heavily on LLM-synthetic labels (which may introduce model bias) or pure human annotation (which is difficult to scale), SciNet leverages objective citation network topology grounded in established scientometrics (e.g., Disruption Index, Uzzi's Z-score). We further cross-validate these signals with expert human annotations to ensure that the ground truth reflects genuine scientific impact and logical evolution.

| Dataset (Benchmark) | Corpus Scale | Task Focus | Supervision Signal |
|---|---|---|---|
| LitSearch (2024) | 64,183 papers | Semantic Retrieval / QA | LLM Generation + Human Annotation |
| PASA (2025) | ~7,000 papers (ICLR 2023, ICML 2023, NeurIPS 2023, ACL 2024, CVPR 2024) | Structural Information Extraction | LLM Synthetic Labels |
| STARK (2024) | 3,037,885 nodes | Entity-centric Relational Hops | Knowledge Base Extraction + LLM |
| **SciNet (Ours)** | **269,091,010 papers** | **Relation-aware Reasoning** | **Objective Topology & Scientometrics + Human Validation** |

*Table 5.* Comparison between SciNet and existing scientific retrieval benchmarks.

## A.2. Experimental Details

For both SciBERT and the Qwen3-Embedding model, for each target paper under consideration, we concatenate the paper's title and abstract into a single string in the format "title: <title>, abstract: <abstract>". We then obtain embeddings for the concatenated string using the respective embedding model. The embedding dimensionality is 768 for SciBERT and 4,096 for Qwen3-Embed.

To enable efficient large-scale similarity search over these high-dimensional embeddings, we constructed a dedicated index using the Faiss library. The preprocessing pipeline first performs L2 normalization on all embeddings, which ensures that maximum inner-product search is equivalent to finding the highest cosine similarity, a standard metric for semantic relevance. We employ an IndexIVFPQ structure, which combines an inverted file system (IVF) for coarse partitioning of the search space and product quantization (PQ) for compact vector representation. Specifically, the algorithm first partitions the entire vector space into 16,384 cells using k-means clustering, where each cell is represented by a centroid vector. This IVF structure enables a substantial pruning of the search space by only examining a small subset of cells closest to the query vector. Within each cell, vectors are further compressed using product quantization: each vector is split into 32 sub-vectors, and each sub-vector is quantized into an 8-bit code pointing to the nearest centroid in a learned codebook.

This two-level scheme, coarse quantization via IVF followed by fine-grained PQ, significantly reduces memory footprint

while accelerating retrieval, achieving a favorable trade-off between search accuracy and efficiency. All embeddings and queries are computed on a single NVIDIA A100 GPU.

For PASA and PaperQA2, we strictly followed the implementations provided in their respective GitHub repositories. PASA was deployed on a local A100 GPU using its pretrained pasa-7b-crawler and pasa-7b-selector models, with API keys configured for Google Search and other relevant tools. For PaperQA2, we performed segmentation and embedding on all papers to fully leverage the model's capabilities. For all OpenAI models, we accessed them using official API keys from the OpenAI platform.

### A.3. Practical Deployment Costs of Scientific Networks

In fact, with a properly designed structured indexing architecture, the computational cost of relation-aware retrieval over large-scale scientific networks is remarkably low. Taking our actual infrastructure as an example: we constructed a local hierarchical relational index for the 269-million-paper citation network using only a lightweight SQLite database. This system operates efficiently in a pure 24-core CPU environment, controlling the single-query latency for relational searches within the massive network at the millisecond level.

For real-world industrial environments, further integrating enterprise-grade database systems such as MySQL or Elasticsearch, or introducing GPUs to accelerate retrieval, would improve throughput and speed by orders of magnitude. Therefore, deploying relation-aware retrieval on large-scale scientific corpora is entirely feasible from an engineering perspective and highly cost-effective.

### A.4. Limitations

While SciNet provides a large-scale and rigorous benchmark for relation-aware scientific literature retrieval, we acknowledge several inherent limitations that warrant further discussion and provide directions for future work.

First, the citation networks utilized in SciNet may inherit systemic biases prevalent in scholarly communication, such as the *Matthew Effect*. High-profile classic papers or those from prestigious institutions often occupy central positions in the citation graph. While our core scientometric indicators, such as the Disruptiveness Index, are mathematically designed to be scale-independent by normalizing for total citation volume, tasks involving path reconstruction are inherently more sensitive to network density. Specifically, the high topological density of influential research can grant a structural advantage during BFS-based path discovery or graph-based searches.

Second, structural noise from automated pipelines remains a challenge. Although we employ state-of-the-art tools like GROBID for PDF parsing and reference extraction, some level of noise is inevitable. When processing early scanned documents or non-standard layouts, OCR errors and paragraph segmentation inaccuracies can occur. Such noise introduces a minor but objective margin of error in evidence extraction for pair-wise sentiment classification or in-text citation anchoring. We aim to mitigate this in future versions by incorporating more robust multimodal parsing models.

Furthermore, there are limitations in our current representation of scientific relationships. Our framework primarily relies on explicit citation links as the ground truth for scientific relationships. However, scholarly influence is not always manifested through formal citations; for instance, parallel independent discoveries or implicit conceptual inspirations may exist without direct graphical edges. By focusing on strong graph connectivity, SciNet may overlook these "implicit" scientific impacts. Future iterations could integrate co-authorship networks and semantic concept evolution to provide a more holistic view of scientific discovery.

Finally, the current query set lacks sufficient linguistic randomness and conversational diversity. Our queries are primarily standardized test queries derived from structured rules to ensure evaluation objectivity. While this covers 2,640 fine-grained subfields, it does not fully capture the diverse and sometimes unpredictable nature of real-world user queries. Future updates will focus on incorporating authentic "wild queries" from academic search logs and utilizing LLMs for query paraphrasing to better mirror real-world application scenarios.

### A.5. Detailed Dataset Construction Pipeline

To ensure the scalability, rigor, and reliability of SciNet, our data curation methodology is formalized into three paradigms:

1. **Automated:** Leveraging scripts and workflows to process large-scale data efficiently, ensuring dataset scalability.

2. **Rule-based (Theoretically Grounded):** Computing the ground truth using authoritative theoretical research, rigorous mathematical formulations, objective graph-theoretic algorithms, and established scientometric principles.
3. **Manual:** Introducing human expert intervention at critical logical nodes, applying blind testing and noise filtering to strictly control dataset quality.

The complete construction pipeline and the corresponding quality assurance mechanisms are summarized in Table 6.

| Pipeline Stage | Description | Curation Method | Quality Assurance Mechanism |
|---|---|---|---|
| **1. Database Acquisition** | Acquired the complete OpenAlex snapshot of 269 million metadata records and integrated the arXiv PDF corpus. | **Automated** | Directly utilized official full snapshots (OpenAlex RELEASE 2025-07-07 and arXiv official full database as of July 7, 2025) to ensure data integrity. |
| **2. Subfield Taxonomy Curation** | Filtered out 2,640 fine-grained scientific subfields based on the topic classification system. | **Automated + Manual** | Following automated preliminary classification, human experts conducted manual review and aggregation to ensure subfields are representative and comprehensively covered. |
| **3. Ego-centric Task Construction** | Automatically generated attribute queries based on subfield classifications; calculated paper attributes to construct the ground truth. | **Rule-based** | Based on authoritative index calculation formulas (e.g., $D_i$), grounded in established scientometric theory to ensure objective rigor. |
| **4. Pair-wise Task Construction** | Utilized GROBID to extract same-paragraph co-occurrence evidence, and used LLMs to classify citation context sentiment. | **Automated + Manual** | Human experts blindly annotated 200 citation sentiments, achieving 98% agreement with LLM outputs (only minor boundary discrepancies), demonstrating high label reliability. |
| **5. Path-wise Task Construction** | Ran BFS to identify maximum cumulative citation paths between classic and frontier papers across decades. | **Rule-based + Manual** | *Algorithmic Exhaustion:* BFS traversal combined with collective credit allocation reflects community consensus. *Human Inspection:* Experts manually verified the constructed paths to eliminate meaningless links lacking scientific coherence. |

*Table 6.* The construction pipeline and quality assurance mechanisms of SciNet.

As shown in the pipeline, SciNet deeply integrates automated computation, rule-based derivation, and targeted human validation. By confining LLM usage to tightly controlled annotation tasks and grounding the core relational logic in established scientific rules, we ensure the dataset's reliability while avoiding over-reliance on LLMs.

## B. Supplementary Experimental Results

### B.1. Case Study 1: Identifying Structural Disruption in Direct Preference Optimization

To demonstrate the limitations of current flagship retrieval systems in capturing deep scholarly relations, we conducted an in-depth case study focused on the query: *"What are the top 5 most disruptive papers in the field of Direct Preference Optimization (DPO)?"* Identification of disruption requires a system to move beyond keyword matching to decode the citation network's evolution, specifically identifying works that fundamentally shift the research trajectory. We compared the outputs of four representative retrieval paradigms against the scientometric ground truth defined by SciNet, as summarized in Table 7.

The failure of existing models to accurately retrieve disruptive papers stems from a fundamental inability to distinguish between *semantic relevance* and *topological significance*. As observed in the results from dense embedding models like Qwen-Embedding, the reliance on static representations leads to extreme "semantic drift," where the system retrieves classical decision theory papers from the 1950s simply because they share the tokens "Preference" and "Optimization." This underscores the incapacity of content-only retrieval to decode the discursive landscape of modern AI. Furthermore, search-enabled agents like GPT-4o-search and PASA exhibit a severe "recency bias," prioritizing 2025 ArXiv uploads such as *Linear Preference Optimization*. Although these publications exhibit lexicographical relevance, they are functionally

| Model Category | Top-5 Retrieved Papers (Representative) | Analysis of Failure Mode |
|---|---|---|
| **GPT-4o-search** | Linear Preference Optimization (2025), In-context Ranking (2025), BPO (2025) | **Recency Bias:** Conflates temporal proximity with structural impact; retrieves unproven 2025 preprints with zero citation impact. |
| **o3-deep-research** | Multi-Turn DPO (2024), New Desiderata for DPO (2024), Active Learning for DPO (2025) | **Incremental Focus:** Recalls task-specific variants that maintain the status quo rather than papers that reshape the DPO paradigm. |
| **Qwen-Embedding** | Lexicographic orders (1953), Creating optimal objects (2001) | **Semantic Drift:** Superficial keyword matching across disconnected domains; fails to localize the specific AI alignment context. |
| **PASA (2025)** | Risk-aware DPO (2025), SGDPO (2025), 2D-DPO (2024), ICDPO (2024) | **Relation-Blindness:** Provides an exhaustive list of DPO-tagged papers without the ability to rank them by topological significance ($D_i$). |

*Table 7.* Retrieval Results Comparison for DPO Disruptiveness Identification.

incremental; by offering only minor hyper-parameter adjustments or scheduling tweaks, they fail to produce the structural disruption required to marginalize prior research. Current agents prioritize the retrieval of similar content, whereas genuine scientific discovery necessitates identifying atypical combinations that fundamentally advance a field.

The depth of this failure is most apparent in the omission of key structural milestones that redefined the DPO evolutionary path. For instance, none of the models successfully retrieved *KTO: Model Alignment as Prospect Theoretic Optimization* (Ethayarajh et al., 2024), which disrupted the field by demonstrating that alignment can be achieved through binary signals rather than paired preferences, representing a total paradigm shift in data requirements. Similarly missed were *IPO: Identity Policy Optimization* (Azar et al., 2023), which theoretically resolved the overfitting instabilities of the original DPO, and *Diffusion-DPO* (Wallace et al., 2023), which cross-pollinated DPO into generative vision. Instead, models prioritized papers with high title similarity that remain entirely within the original DPO's shadow. This discrepancy highlights that providing agents with relational ground truth, such as citation sentiment and network disruption indices ($D_i$), is not merely an enhancement but a necessity for meaningful literature synthesis. While flagship models showed a recall of less than 20%, a relation-aware approach correctly identifies the seminal *Direct Preference Optimization* (Rafailov et al., 2023) and its most influential theoretical successors, bridging the gap between shallow search and deep scientific reasoning.

### B.2. Case Study 2: Reconstructing Evolutionary Trajectories in General Vision Models

To evaluate the capacity of retrieval agents in capturing the long-range collective dynamics of scientific progress, we designed a path-wise retrieval task: *"What is the most influential citation path from 'Neocognitron' (1980) to 'Swin Transformer' (2021) in the field of General Vision Models?"* This task requires the system to not only identify relevant papers but to reconstruct the causal and citational lineage that connects classical bio-inspired pattern recognition to modern hierarchical transformers. The comparative performance of leading models against the ground truth trajectory is detailed in Table 8.

The primary challenge in path-wise retrieval lies in maintaining both thematic consistency and citational connectivity across decades of research. As observed in the results from Qwen-Embedding and PASA, traditional retrieval paradigms suffer from a complete inability to capture sequential dependencies. Embedding models prioritize surface-level semantic similarity to the query's keywords, which results in a cluster of modern Transformer variants that ignore the historical starting point of the Neocognitron. Similarly, PASA fails to move beyond "topic localization" by retrieving contemporary surveys that discuss vision models but do not provide the explicit multi-step evidence chains required to reconstruct a developmental trajectory. These models treat scientific literature as an unordered collection of documents rather than a semantically enriched graph of evolving ideas, thereby failing to decode the underlying relational dynamics.

While reasoning-heavy agents like o3-deep-research produce a logically sound story of computer vision by tracing the shift from CNNs to Transformers, they often fail the strict connectivity and consistency metrics when compared to the high-influence ground truth path. The ground truth trajectory, identified through a breadth-first search (BFS) on the citation network, reveals a dense lineage of architectural milestones including *Deep Residual Learning* (ResNet), *Faster R-CNN*, *VGGNet*, and *Rich Feature Hierarchies* (R-CNN). These papers represent the collective credit allocation of the scientific community; they serve as the actual hubs that facilitated the transition from shift-invariant neural networks to hierarchical

| Model Category | Retrieved Path / Results (Representative) | Analysis of Failure Mode |
|---|---|---|
| o3-deep-research | Neocognitron → LeNet → AlexNet → ResNet → ViT → Swin Transformer | **Structurally Plausible but Topologically Incomplete:** Constructs a coherent CNN-to-Transformer narrative, yet omits high-centrality intermediate hubs (e.g., Faster R-CNN), leading to an under-connected evolution graph. |
| GPT-4o | Neocognitron → ViT → Swin Transformer | **Severely Fragmented Trajectory:** Collapses decades of architectural evolution into sparse jumps, failing to model the continuous refinement process across convolutional paradigms. |
| Qwen-Embedding | LocalViT, Scaling ViT, Multiscale ViT | **Endpoint-Centric Retrieval:** Focuses on semantically similar works near the target models, but fails to recover a temporally ordered developmental trajectory. |
| PASA (2025) | A Survey of Vision-Language Pre-training, Survey on Vision Autoregressive Models | **Keyword-Driven Misalignment:** Retrieves recent survey papers with lexical overlap, rather than reconstructing the underlying citation or innovation pathway. |

*Table 8.* Path-wise Retrieval Comparison for Vision Model Evolution.

transformers. Flagship models consistently bypass these essential milestones, particularly the critical object detection frameworks that pioneered the use of region-based convolutional hierarchies. This reveals a fundamental weakness in capturing the collective dynamics of scientific progress. This discrepancy underscores that reconstructing intellectual lineages requires relation-aware retrieval capable of modeling temporal progression and causal reasoning, pushing beyond shallow semantic matching toward genuine knowledge synthesis.

## C. Prompts

The LLM prompts for novelty evaluation in the Ego-Centric task are as follows:

```
You are an expert academic reviewer. Your task is to evaluate a scientific paper on
    its Novelty.

### Definition of Novelty:
Definition: Novelty refers to the uniqueness and originality of the research
    question, methodology, data, or conclusions relative to existing research.
Focus: Does the paper introduce new ideas, perspectives, or methods within the
    existing body of knowledge? For example, applying a method from Field A to Field
    B for the first time.
Scoring Criteria: A score of 0 represents completely derivative work, while a score
    of 10 represents a highly original and groundbreaking idea.

---

Please evaluate the Novelty of the following paper based on the definition provided.

**Title**: [Paper Title]

**Abstract**: [Paper Abstract]
(or: [No abstract provided. Please evaluate based on the title alone.])

---
```

```
Your response MUST be a single JSON object with 'score' (an integer from 0 to 10)
    and 'reasoning' (a brief explanation).
```

The LLM prompts for disruptiveness evaluation in the Ego-Centric task are as follows:

```
You are an expert academic reviewer. Your task is to evaluate a scientific paper on
    its Disruptiveness.

### Definition of Disruptiveness:
Definition: Disruptiveness refers to the way a paper influences subsequent research-
    does it cause future work to cite the paper itself, rather than the previous
    works it was built upon?
Focus: Does the paper change the direction of a research field or its methodologies,
     causing prior work to be marginalized? For example, the foundational papers on
    CRISPR gene-editing technology opened new research avenues and made previous
    editing methods obsolete.
Scoring Criteria: A score of 0 represents no disruptive potential (e.g., a review
    paper), while a score of 10 represents the potential to highly transform a field.

---

Please evaluate the Disruptiveness of the following paper based on the definition
    provided.

Title: [Paper Title]

Abstract: [Paper Abstract]
(or: [No abstract provided. Please evaluate based on the title alone.])

---

Your response MUST be a single JSON object with:
- "score": an integer from 0 to 10
- "reasoning": a brief explanation supporting the score
```

In the Pair-Wise task, the code for classifying the sentiment tendency of citation context using LLM is as follows:

```
You are an expert in academic literature analysis. Your task is to classify the
    sentiment of a citation context.

Please analyze the following citation context from a research paper, which mentions
    the target paper titled "[Target Paper Title]".

Classify the context into one of three categories:
- Positive: The citing paper praises, builds upon, or confirms the findings of the
    target paper.
- Negative: The citing paper criticizes, questions, or points out limitations of the
     target paper.
- Neutral: The citing paper simply mentions or describes the target paper as
    background information without expressing a strong opinion.

Your response MUST BE only ONE of the three category names: Positive, Negative, or
    Neutral.

Context to analyze:
---
```

```
[Insert citation context here]
---
```

The prompts for LLM evaluation in the Path-Wise task are as follows:

```
You are an expert in scientometrics and academic research. Your task is to evaluate
    the quality of a proposed citation path based on its technical evolution.

### Core Task:
Assess if the provided sequence of papers represents a logical and meaningful
    technological or conceptual evolution from the start paper to the end paper.

### What constitutes a good technical evolution path? (Key Principles)
1. Thematic Consistency: All papers must strictly revolve around the same core
    research topic defined by the query. Deviations into unrelated subjects indicate
    a poor path.
2. Content Cohesion & Logical Flow: The content of adjacent papers must be closely
    related. Each paper should logically follow from the previous one, building upon
    its ideas, refining its methods, or addressing its limitations.
3. Progressive Development: The path must demonstrate clear progress. Later papers
    should represent advancements, extensions, or significant new applications of the
    concepts introduced in earlier papers. The path should tell a story of
    innovation.
4. Represents a Main Line of Inquiry: The path should follow a significant and
    recognized line of development within the research field, not an obscure or
    tangential branch.

### Scoring Criteria (0-10):
- Score 9-10 (Excellent): A perfect or near-perfect path. It is thematically
    consistent, shows clear progressive development, and represents a major line of
    inquiry. The logical flow is impeccable.
- Score 7-8 (Good): A strong, coherent path. Most papers are relevant and show
    progression, but there might be a minor logical gap or a less influential paper
    included.
- Score 4-6 (Mediocre): The path has some relevance but lacks strong cohesion. It
    may include several tangential papers, the logical progression is weak, or it
    fails to capture the main developmental thread.
- Score 1-3 (Poor): The path is largely incoherent. Papers are thematically
    disconnected, show no clear progress, or are mostly irrelevant to the query.
- Score 0 (Failure): A completely random collection of papers with no logical or
    thematic connection.

---

Please evaluate the following citation path based on the detailed criteria provided.

Original Request: [original_query]

--- Proposed Citation Path ---
[Paper list will be inserted here]
-----------------------------

Your response MUST be a single JSON object with 'score' (an integer from 0 to 10)
    and 'reasoning' (a detailed explanation for your score, critiquing the path based
    on the four key principles).
```

In the survey generation experiment 6, the prompts for generating the survey are as follows:

```
You are a helpful academic assistant that generates surveys using retrieved
    literature.

Please generate a survey report in Markdown format based on the following
    information:

Domain: [Domain Name]

Start paper:
Title: "[Start Paper Title]"
Abstract: [Start Paper Abstract]

End paper:
Title: "[End Paper Title]"
Abstract: [End Paper Abstract]

Task:
Generate a survey report describing the technological evolution from the start paper
    to the end paper.
Include key technical developments, major milestones, and method evolution.
Organize the report in a clear Markdown format.

Important:
- Do not use any ground-truth paths.
- Rely only on information retrieved via search capabilities.
```

