# OpenReview forum: "SciNet: Evaluating AI Agents in Relation-Aware Scientific Literature Retrieval"
_ICML.cc/2026/Conference — ICML 2026 regular_

### Official Review · Reviewer_ebAW · 2026-02-18

**Soundness:** 2
**Presentation:** 3
**Significance:** 3
**Originality:** 3
**Overall Recommendation:** 5
**Confidence:** 2

**Summary:**

The paper proposes a new dataset called ScitNet which is a relation-aware retrieval dataset in literature review. The authors start with the discussion on the lack of relations-aware in the modern datasets for literature retrieval.

The proposed relation-aware dataset includes 269 millions papers and consists of three categories: ego-centric, pair-wise and path-wise.  In particular, the ego-centric aims to get the most groundbreaking paper in a field. Pair-wise tries to find a pair of papers that are related to each other. Path-wise targets on the evolution of a topic over a sequence of papers.

The authors also evaluate the datasets using eight different models covering three types of paradigm, namely embedding-based retrieval, agentic retrieval and deep research retrieval. Through the experiments, the authors quantify the limitations of existing methods in the selected tasks.

**Compliance With Llm Reviewing Policy:**

Affirmed.

**Ethical Review Concerns:**

No ethical review concerns are raised.

**Final Justification:**

Overall, I found this work introduces interesting datasets that worth of exploring in the future. Therefore, I maintain my score accordingly.

**Key Questions For Authors:**

Not applicable.

**Limitations:**

Not applicable.

**Strengths And Weaknesses:**

Overall, I found the paper important in a sense that it can contribute to the existing data pool for RAG task in general. However, I am not an expert in the field of RAG related topics. Thus I cannot evaluate the significance of such a dataset. The presentation is clear and I can easily follow through.

---

> ### Author Rebuttal · Authors · 2026-03-31
>
> We sincerely thank you for your positive feedback and for recognizing the clarity and importance of our paper. To provide more context and further address your thoughts on the significance of our dataset, we would like to briefly elaborate on the intuition and domain background behind SciNet.
>
> **Background: Why is traditional RAG insufficient for scientific literature retrieval?** In general RAG tasks (e.g., web Q&A), information retrieval primarily relies on "semantic similarity" or "keyword matching". However, scientific literature retrieval is fundamentally different: scientific knowledge is not a collection of isolated, flat texts, but rather a complex relational network intertwined by citations, peer assessments, refutations, and methodological evolutions. Merely finding a paper with similar keywords does not mean finding the "correct" paper within a scientific context.
>
> Driven by this intuition, SciNet introduces a critical paradigm shift. Its profound significance within the field is primarily reflected in the following three aspects:
>
> - **Filling a fundamental gap in existing benchmarks:** Current mainstream RAG literature retrieval benchmarks (such as LitSearch or PASA) are largely confined to "topical localization" or semantic-level matching. They are unable to comprehend the complex logical lineages between scientific papers. SciNet is the _first_ evaluation benchmark specifically designed for large-scale scientific network relational structures (ranging from ego-centric properties to complex evolutionary paths). It forces AI agents to move beyond mere "keyword matching" to truly understand the trajectory of scientific development.
>
> - **Revealing the "blind spots" of flagship AI Agents:** Through our evaluation, the community can quantitatively observe a harsh reality for the first time: even the most advanced deep research agents, such as `o3-deep-research`, frequently exhibit accuracy rates below 20% when handling relation-aware tasks like "identifying the most disruptive papers" or "reconstructing technological evolutionary paths". SciNet explicitly pinpoints the critical technical bottlenecks that next-generation RAG systems urgently need to resolve.
>
> - **Driving significant practical downstream benefits:** Introducing a relation-aware perspective into retrieval yields tangible performance gains. Taking automatic literature review generation as an example, our experiments demonstrate that after incorporating relation-aware retrieval, the overall quality of AI-generated reviews not only increased by 25.3% , but more importantly, achieved significant improvements in highly challenging, deeper dimensions such as **"Completeness," "Depth," and "Logical Consistency"**. While traditional search agents often merely pile up fragmented content , systems grounded in SciNet's relational links can synthesize intellectual lineage analyses with genuine academic insight. Furthermore, this capability to capture scientific network dynamics can directly empower real-world scientific workflows in the future, such as automated experiment design, innovation ideation, and broader scientific knowledge discovery. This directly proves the immense foundational value of this dataset for developing future "AI Scientists".
>
> We are deeply grateful for your support of our work. We hope that the supplementary background information provided above will further solidify your confidence in SciNet. Thank you once again for your time and your highly valuable review!

---

> > ### Author Rebuttal · Reviewer_ebAW · 2026-03-31
> >
> > I thank the authors for sharing the comments with me. Overall, I found the work is valuable. Thus, I maintain my score.

---

### Official Review · Reviewer_UsMY · 2026-03-12

**Soundness:** 2
**Presentation:** 1
**Significance:** 3
**Originality:** 2
**Overall Recommendation:** 3
**Confidence:** 3

**Summary:**

The paper introduces SciNet, a large scale benchmark designed to evaluate AI systems on tasks related to scientific literature exploration and relational reasoning over scholarly data. The dataset is constructed from a large corpus of scientific publications and includes tasks that capture different aspects of scientific knowledge discovery, such as citation retrieval, pairwise relations between papers, and path based connections in citation networks. The benchmark evaluates a diverse set of systems, including embedding-based models, agentic retrieval systems, and deep research agents, in order to study the ability of these architectures to traverse and reason over scientific knowledge graphs. The authors also demonstrate the practical applicability of the benchmark through an automated literature review experiment.

**Compliance With Llm Reviewing Policy:**

Affirmed.

**Final Justification:**

Thanks for addressing some of the shortcomings i have raised. I still think the clarity of the paper can be improved by structuring it more clearly and having an overview of different steps. I also think the authors should make it more clear how their work relates to existing benchmarks, and which concrete shortcomings these benchmarks present that their work tackles, I think the table in figure https://anonymous.4open.science/r/SciNet/Rebuttal/Comparison.png is a great initial step to achieve this. I encourage the authors to extend this table with additional benchmarks, also including more axis (numeric and qualitative) to compare, showcasing the advantage of your dataset.

Thanks a lot for recognizing the shortcomings and tackling them in your rebuttal. I also think the work has a lot of potential. The fact that the recall scores are so low means the dataset indeed reveals shortcomings in existing models and opens new avenues to be explored.

I increased my score.

**Key Questions For Authors:**

1. What parts of the dataset creation pipeline involve manual validation?
2. What exact subset of the OpenAlex corpus is included in SciNet? Are all papers indexed, or only those belonging to certain topics or time ranges?
3. How does SciNet compare to existing scientific retrieval or citation datasets in terms of scale, tasks, and supervision signals? Could the authors include a comparative table?
4. Could the authors more formally specify the tasks, including what information is available during inference and what inputs are provided to the models?

**Limitations:**

No limitations discussed

**Strengths And Weaknesses:**

Strength:
1. SciNet benchmark is constructed from a very large scientific database containing approximately 269 million papers across multiple disciplines, which allows this benchmark to reflect the complexity and scale of real- world scientific literature. Additionally, this scale enables the evaluation of retrieval systems under conditions that more closely resemble practical workflows.
2. The dataset is designed to evaluate retrieval systems not only on semantic similarity but also on their ability to capture relational structures within scientific literature, such as relationships between papers, citation dynamics, and the evolution of research topics. This highlights an important limitation of current literature retrieval systems that mainly rely on embedding -based similarity to retrieve related literature.
3. The evaluation includes a broad set of systems spanning three categories : embedding models, agentic retrieval systems, and deep research agents. This diverse setup provides useful insight into how different architectural paradigms perform on relation-aware scientific retrieval tasks.
4. The benchmark introduces different task types, including ego-centric retrieval, pairwise relation identification, and path-based reconstruction of scientific development, which attempt to capture different forms of relational reasoning over scientific literature.
5. The authors include an experiment where agents use SciNet to generate literature reviews, showing how relation-aware retrieval can improve downstream research assistance workflows. This experiment helps to illustrate the potential practical value of the dataset.

Weaknesses:
1. I think that the motivation for introducing SciNet in the abstract and introduction is rather unclear. For example, in abstract, the paper initially discusses the limitations of AI agents for literature retrieval, and then introduces a dataset as a solution. In my view, since the main contribution of the paper is a benchmark dataset, the motivation would be clearer if it first focused on the limitations of existing datasets and evaluation benchmarks, and explicitly explained what aspects of scientific reasoning they fail to capture.
2. The introduction refers to a "manually validated corpus" (line 119), yet the dataset appears to be constructed primarily through automated pipelines over a corpus of approximately 269 million papers. Therefore, it is unclear to me what part of the dataset creation process involves manual validation. This statement should be clarified to avoid potential over-claiming, and maybe provide a table concretely outlining which steps are automatic and which are manual and in case of automatic how the authors ensure the annotation quality.
3. Related to previous point, some tasks rely on labels generated automatically using external models (e.g., GPT-based models for citation sentiment classification). I think this raises the concern about the reliability of the evaluation, particularly when similar LLM-based systems are also among the evaluated models. I think the paper would benefit from a stronger evaluation including manual annotations for at least a subset of the test set to provide an independent ground truth.
4. The way I see it, the proposed Cite-Acc metric may penalize systems that retrieve citations that are correct and relevant but not included in the reference list of the original paper. Because scientific papers typically cite only a subset of relevant literature, I think this metric may underestimate system performance. I would suggest including complementary metrics such as recall-based or relevance-based ones, that could provide a more complete evaluation of retrieval quality.
5. Related to the weakness in point 3., I also observe that several dataset labels seem to be generated automatically using external models or heuristics. For example, GPT-based models are used to produce certain labels (e.g., lines 214–217 and 256). If similar GPT-based systems are also evaluated in the benchmark, this may introduce evaluation bias. Additionally, some tasks rely on labels that can be generated automatically using old tools such as GROBID, raising the question of what additional insight is gained by evaluating newer and more advanced LLM-based systems on the same automatically generated labels. Finally, certain ground-truth relations are constructed using BFS over citation networks. While computationally straightforward, this approach seems to me  simplistic, and it is unclear to me how well such automatically derived paths correlate with meaningful relationships between papers from a human perspective.
6. I think the paper does not always clearly specify: what information is available during dataset construction, what information is available to models at inference time, and what the exact input-output format of each task is. Providing a clearer formalization (e.g., using concrete and consistent notation to represent each of the components in the dataset, and their use during construction/evaluation) of the tasks would improve reproducibility and interpretability of the results.
7. The paper does not include a discussion of the limitations of the dataset and evaluation framework, such as potential biases in the citation network, noise introduced by automatic labeling, or limitations of the relational definitions used. A dedicated limitations section would improve the completeness of the work.

---

> ### Author Rebuttal · Authors · 2026-03-31
>
> We greatly appreciate your insightful comments. Below, we address each of your concerns in detail.
>
> **Manual Validation (Weakness 2 & Question 1)**
> We have added a detailed table outlining our data construction pipeline, which can be viewed here:
>
> https://anonymous.4open.science/r/SciNet/Rebuttal/Pipeline-table.png
>
> By confining LLM usage to tightly controlled annotation tasks and grounding the core relational logic in established scientific rules, **we ensure the dataset’s reliability while avoiding over-reliance on LLMs**.
>
> **Label Reliability (Weakness 3)**
> As shown in the table above, LLM annotation is only used for Pair-wise citation sentiment classification. To enhance the credibility, we have added an experiment: human experts manually annotated the sentiment of 200 citation contexts and compared them with LLM results, achieving a **98% agreement rate** (with only 4 minor discrepancies on the "neutral/positive" boundary). This strong alignment proves the high reliability of the LLM annotations.
>
> **Validity of the `Cite-Acc` Metric (Weakness 4)**
> If a model retrieves a highly relevant paper that was not actually referenced by the original paper, it is indeed penalized under the current `Cite-Acc` metric. Following your suggestion, we have added an "Auxiliary Semantic-Relevance (`Aux-Relevance`)" metric for a fairer evaluation. We utilized GPT-5 to perform binary relevance scoring on the retrieved titles & abstracts. The results are provided here:
>
> https://anonymous.4open.science/r/SciNet/Rebuttal/Aux-Relevance.png
>
>
> **Clarifications on Evaluation Bias, Automatic Labeling, and Path Validity (Weakness 5)**
> - **Potential LLM-induced bias.**  For LLM-as-a-judge metrics (Novelty-LLM, Disruption-LLM), we have added a small-scale human alignment study. Two AI Ph.D. annotators performed double-blind novelty scoring on 50 samples per model, yielding a strong correlation with LLM judgments (Pearson _r_ = 0.83), with no statistically significant bias favoring GPT (e.g., _gpt-4o-search_).
>   For LLM-generated labels (citation sentiment), we have also performed manual validation, achieving 98% human-LLM agreement over 200 samples.
> - **Automatic labeling.** GROBID is only used for reference parsing, not for generating labels. The final labels are constructed via multi-stage pipelines combining structured extraction, semantic annotation, and manual verification.
>   Moreover, there is a fundamental asymmetry between dataset construction (open-book) and model evaluation (closed-book). During inference, agents cannot access parsing tools or ground-truth documents; instead, they must autonomously retrieve and cross-validate relations from unstructured web-scale corpora. Therefore, these tasks evaluate relational reasoning and information synthesis, rather than simple parsing capabilities.
> - **Validity of citation paths.**  Path construction is a pipeline combining theoretical grounding, algorithmic filtering, and human validation. Specifically, it is inspired by collective credit allocation, prioritizing paths through high-impact hub papers (via cumulative citation signals). Candidate paths are then filtered by expert checks to remove semantically disconnected links. Finally, we include a human-evaluated *Rationality-Human* metric, where blind expert assessment shows strong alignment, supporting the meaningfulness of the constructed trajectories.
>
>
> **Supplementary Formal Definitions (Weakness 6 & Question 4)**
> We will add unified mathematical formalizations in the revision to explicitly clarify information availability. Let $\mathcal{C}$ be the corpus metadata and $\mathcal{G}=(\mathcal{V}, \mathcal{E})$ be the underlying citation graph.
> - **Inference Phase (Input/Output):** The graph $\mathcal{G}$ is strictly hidden. Models only receive a query $q$ and access to $\mathcal{C}$ (web agents map search results back to $\mathcal{C}$). The output is a ranked list of $K$ papers: $\hat{Y} = M(q | \mathcal{C}) = [p_1, \dots, p_K]$, where $p_i \in \mathcal{C}$.
> - **Construction Phase (Ground Truth):** The true labels $\mathcal{Y}^*$ are computed using the full graph $\mathcal{G}$. For the Ego-centric disruptiveness task, we maximize the Disruptiveness Index over a topic subset $\mathcal{C}_{\text{topic}}$:
>
> $$
> \begin{aligned}
> \mathcal{Y}^*
> &= \arg\max_{P} \sum_{p \in P}
> \frac{N_I(p) - N_J(p)}{N_I(p) + N_J(p)}, \\
> &\text{subject to } P \subset \mathcal{C}_{\text{topic}}, \ |P| = K,
> \end{aligned}
> $$
> where $N_I(p)$ and $N_J(p)$ are topological state variables derived entirely from the edge set $\mathcal{E}$, representing subsequent citations that respectively bypass or include $p$'s prior references.
>
>
> **OpenAlex Corpus (Question 2)**
> We actually index all papers in the OpenAlex dataset.
>
> **Comprehensive Comparison (Question 3)**
> We add the following comparison table, contrasting SciNet with existing benchmarks across three dimensions:
> https://anonymous.4open.science/r/SciNet/Rebuttal/Comparison.png

---

> > ### Author Rebuttal · Reviewer_UsMY · 2026-04-01
> >
> > Thanks a lot for your complete answer, for recognizing the weaknesses of your approach, and for making it stronger and addressing my suggestions. I would also like to see the recall metric on the citations, what fraction of ground truth citations are covered by each method?

---

> > > ### Author Response · Authors · 2026-04-01
> > >
> > > We sincerely thank you for the positive feedback and the constructive engagement.
> > >
> > > Following your suggestion, we have computed the Recall@20 and Recall@50 metrics for all 8 systems to measure the fraction of ground-truth citations covered. For each query, this metric is formally calculated as $\frac{|R_K \cap G|}{|G|}$, where $R_K$ is the set of top $K$ retrieved papers and $G$ is the full ground-truth set. The results averaged over all queries are summarized below:
> > >
> > > | Category     | Models                | Recall@20 | Recall@50 |
> > > | ------------ | --------------------- | --------- | --------- |
> > > | Embedding    | SciBERT               | 0.152%    | 0.326%    |
> > > |              | Qwen3-Embed           | 0.923%    | 1.935%    |
> > > | Agentic      | PaSa                  | 2.856%    | 4.581%    |
> > > |              | PaperQA               | 1.074%    | 2.159%    |
> > > |              | gpt-4o-mini-search    | 8.241%    | 13.380%   |
> > > |              | gpt-4o-search         | 8.917%    | 14.294%   |
> > > | DeepResearch | o4-mini-deep-research | 11.452%   | 17.993%   |
> > > |              | o3-deep-research      | 11.968%   | 18.432%   |
> > > **Observations:**
> > > 1. **Agentic Advantage in Relation-aware Retrieval:** As shown in the table, agentic systems with multi-step reasoning and dynamic retrieval significantly outperform embedding-based methods. This gap highlights that iterative search and reasoning are critical for uncovering a broader portion of the citation network.
> > > 2. **The Enduring Challenge:** Even the best deep research systems achieve less than 20% recall, indicating that existing retrieval methods still have notable limitations in understanding and exploring relationships within the scientific network, leaving substantial room for improvement and optimization.
> > >
> > > We will add these results and the detailed discussion to the revised manuscript to provide a more comprehensive view of retrieval performance.

---

### Official Review · Reviewer_6BhH · 2026-03-12

**Soundness:** 3
**Presentation:** 3
**Significance:** 3
**Originality:** 2
**Overall Recommendation:** 3
**Confidence:** 3

**Summary:**

The paper leverages large-scale open-source academic network data and uses Scientometrics metrics along with LLMs to construct three types of queries and their corresponding retrieval results, namely: (1) ego-centric retrieval of papers with novel knowledge structures, (2) pair-wise identification of scholarly relationships, and (3) path-wise reconstruction of scientific evolutionary trajectories.

**Compliance With Llm Reviewing Policy:**

Affirmed.

**Final Justification:**

Thank you for the rebuttal and the additional clarifications. After considering both the paper and the authors’ response, I maintain my final recommendation.

The paper has clear strengths. The topic is important, the motivation for relation-aware scientific literature retrieval is meaningful, and the work shows promising interdisciplinary integration between scientometrics, retrieval, and LLM-based evaluation. The overall logic is generally clear, and several of the main claims are reasonably supported. These aspects make the paper potentially valuable.

However, I still have concerns about the paper’s originality and empirical justification. In particular, the ego-centric formulation mainly focuses on novelty, which captures only one dimension of realistic retrieval needs. The rebuttal usefully adds a “paper quality” experiment, but I still think the paper would benefit from a clearer justification of why novelty is the central starting point and how the framework extends to broader retrieval intents.

I also appreciate the clarification that the full path-wise benchmark contains 2,100 queries, while the 40-query subset is used only for the downstream literature-review-generation evaluation. This addresses part of my earlier concern. However, I still think the paper needs more concrete and reproducible evidence that this 40-query subset is sufficiently representative, rather than only a high-level description of the sampling strategy.

The rebuttal also constructively addresses presentation issues by adding an illustration for the disruption index and promising stronger discussion of limitations and related work. These are helpful improvements, but in my view they only partially resolve the underlying concerns, especially regarding query diversity and generalizability.

Overall, I think this is a thoughtful and promising paper, but I remain unconvinced that the current version is strong enough in originality, scope of task formulation, and empirical support for some key design choices. The rebuttal improved the paper and clarified several points, but it did not change my overall evaluation. Therefore, I maintain my final recommendation.

**Key Questions For Authors:**

1. What criteria were used to select the 40 queries in the Path-wise dataset, and can the authors justify why this quantity is sufficient for robust evaluation? (weakness 3)
2. The Ego-centric Relation metric only captures novelty. Have the authors considered incorporating additional retrieval dimensions such as paper quality, and what challenges would this entail? (weakness 2)
3. The authors could provide an illustrative figure or example demonstrating the disruption index and its implication. (weakness 1)

**Limitations:**

The paper does not fully discuss its limitations in social impact, and this should be addressed.

**Strengths And Weaknesses:**

Strengths:
1. Most of the claims stated in the paper are well-supported and presented with clear logic.
2. The research topic is important, and the paper provides a reasonable motivation for addressing “Relation-Aware Scientific Literature Retrieval.”
3. The paper leverages Scientometrics metrics to construct datasets, demonstrating the study’s strong interdisciplinary integration.

Weaknesses:
1. For the novelty metrics (especially the disruption index), adding illustrative figures could help. At present, the disruption index may not be immediately intuitive, so visual examples could make it easier for readers to grasp, especially those encountering the metric for the first time.
2. The Ego-centric Relation metric captures only one aspect: novelty. It would be interesting to see exploration of additional aspects of user retrieval needs, such as identifying high-quality papers.
3. The Path-wise dataset includes 40 representative queries, which may be insufficient in quantity. Providing the clear criteria how these 40 queries were selected would help.
4. The dataset has limited generalization in terms of queries. It may not fully capture the randomness and diversity of user queries.
5. The writing and synthesis of the related work section are relatively weak. Strengthening this section would help highlight the paper’s unique contributions.

---

> ### Author Rebuttal · Authors · 2026-03-31
>
> We sincerely thank you for your valuable feedback. We respond to your concerns point-by-point as follows:
>
> **Adding an illustration for the Disruption Index (Weakness 1 & Question 3)**
> Thank you for your valuable suggestion. We have added an intuitive illustration of the Disruption Index, which can be viewed here:
>
> [https://anonymous.4open.science/r/SciNet/Rebuttal/Illustration.png](https://anonymous.4open.science/r/SciNet/Rebuttal/Illustration.png)
>
> This illustration visually contrasts incremental research (where subsequent papers cite both the target and its references) with disruptive research (where subsequent papers cite _only_ the target, bypassing prior foundations to initiate a new paradigm). We will include this illustration in the revised manuscript to enhance overall readability.
>
>
> **Expanding evaluation dimensions for Ego-centric tasks (Weakness 2 & Question 2)**
> Following your suggestion, we have added an experiment that requires the model to retrieve the "highest-quality" papers within each subfield, using citation counts as the indicator of quality. We construct 2,640 new queries (_“What are the highest-quality papers in [Topic X]?”_) and evaluate 8 models. The average citation counts of the top-10 retrieved papers are shown in the following link:
>
> [https://anonymous.4open.science/r/SciNet/Rebuttal/highest-quality.png](https://anonymous.4open.science/r/SciNet/Rebuttal/highest-quality.png)
>
> The results show that advanced deep research systems (e.g., _o3-deep-research_) significantly outperform embedding-based approaches in retrieving highly cited papers. We will include this experiment in the revised manuscript.
>
>
> **Clarifying the scale of the Path-wise dataset (Weakness 3 & Question 1)**
> We would like to clarify a potential misunderstanding regarding dataset scale. Our Path-wise evaluation dataset contains **2,100 queries**, as described in Sections 2.1 and 5.1. All results reported in Table 3 are based on this full set, ensuring robustness and statistical significance.
>
> The **40 queries** you mentioned are a specifically sampled subset used for evaluating the practical downstream application (i.e., automated literature review generation) detailed in Section 6. To ensure this subset is highly representative of the broader dataset, we employed a strict stratified sampling strategy based on three dimensions. First, to prevent domain bias and ensure the generalizability of the evaluation, queries were proportionally sampled across major scientific macro-fields to maintain disciplinary diversity. Second, we selected evolutionary paths with varying historical timeframes, incorporating both rapid short-term developments and long-span historical trajectories. Finally, to account for topological complexity, the subset includes a balanced mix of different ground-truth path lengths and branching
> factors, ensuring that the models are rigorously tested across varying levels of reasoning difficulty.
>
> We limited the sample size to 40 queries primarily to ensure the high quality of the human evaluation labels. Evaluating AI-generated long-form literature reviews is highly cognitively demanding. It requires experts to conduct in-depth reading, perform meticulous fact-checking against the actual citation graph, and execute cross-blind preference ranking. Scaling up such a resource-intensive task would inevitably compromise judgment quality and introduce noise. By concentrating our expert resources on 40 representative cases, we ensure that every human label undergoes rigorous and careful validation.
>
>
> **Generalization of queries (Weakness 4)**
> Thank you for the valuable suggestion. To ensure the objectivity of complex evaluations (such as scientometric indicators) and the reliability of the ground truth, the current queries were primarily extracted as standardized probes based on structured rules. Although we have covered 2,640 fine-grained subfields across 7 major scientific domains in terms of breadth, we acknowledge that there is indeed a lack of diversity and randomness in the query expressions. We will add an in-depth discussion of this issue in a newly added "Limitations" section in the revised manuscript. In future work, we plan to collect user interaction logs (search logs) from real-world academic retrieval systems or "AI research assistants" to extract authentic "wild queries." This will significantly supplement the diverse, real-world retrieval intents stemming from daily academic workflows that are not covered in current dataset.
>
>
> **Enriching the Related Work (Weakness 5)**
> Thank you for the suggestion. In the revised manuscript, we will substantially expand Section 7 to provide a deeper comparative analysis of existing "graph retrieval" methods and "scientific literature retrieval benchmarks." This enhanced synthesis will explicitly contrast prior works with SciNet, clearly highlighting our unique contributions to the evaluation of large-scale "relation-aware" networks.

---

> > ### Author Rebuttal · Reviewer_6BhH · 2026-04-02
> >
> > Thank you for the rebuttal. I appreciate the authors’ effort in clarifying the motivation, the dataset construction process, and the intended scope of the proposed relation-aware retrieval setting. Some of my concerns have been addressed to a certain extent.
> >
> > However, my main concerns are only partially resolved. In particular, I still think the paper would benefit from a clearer justification of the Path-wise dataset design. The current benchmark contains 40 representative queries, and I would like the authors to explain more explicitly how these queries were selected, whether the selection followed predefined criteria rather than manual preference, and what evidence supports that this set provides adequate coverage of different query types, topics, or path complexities. If possible, the paper would be strengthened by reporting more concrete statistics or selection criteria to justify why this quantity is sufficient for drawing robust conclusions.
> >
> > I also remain concerned that the Ego-centric relation formulation currently focuses on only one retrieval dimension, namely novelty. While I understand the authors’ intention to scope the problem, I encourage the authors to discuss more clearly why novelty was prioritized over other practically meaningful retrieval needs, such as paper quality, influence, or broader user-oriented retrieval intents, and what challenges would arise in extending the framework in these directions.
> >
> > In addition, although the rebuttal helps clarify parts of the methodology, I still think the paper would benefit from a more explicit discussion of the dataset’s generalization limits, especially regarding the diversity and randomness of real user queries. Strengthening the related work section would also help better position the paper’s contribution relative to prior literature.
> >
> > Finally, I encourage the authors to include a more intuitive illustration or example for the disruption index in the revision, as this would make the novelty-related metric easier to understand for a broader audience.
> > Overall, I appreciate the clarifications in the rebuttal, but I consider the concerns only partially resolved.

---

> > > ### Author Response · Authors · 2026-04-02
> > >
> > > We sincerely thank the reviewer for your follow-up reply and continued attention to our work. **We would like to respectfully clarify that we have already addressed these concerns in our initial rebuttal, providing new experiments, explanations, and illustrations.** To facilitate your review, we respond to your four questions in order below:
> > >
> > > # 1. Path-wise Dataset Scale and Selection Criteria
> > > There is a misunderstanding regarding the dataset scale. The complete Path-wise evaluation dataset contains **2,100 queries** (the results reported in Table 3 were evaluated on this full set), not 40. The 40 queries you mentioned are strictly a sampled subset used exclusively for evaluating the downstream "automated literature review generation" task (Section 6). To ensure this subset is highly representative, we employed a strict stratified sampling strategy based on three dimensions: (1) proportional sampling across macro-fields (i.e., biology, chemistry, etc.) to maintain disciplinary diversity; (2) coverage of both rapid short-term developments and long-span historical trajectories; and (3) a balanced mix of different topological complexities (e.g., path lengths and branching factors). The sample size was limited to 40 primarily to guarantee the highest annotation quality during the expert human blind evaluation, which is highly cognitively demanding.
> > > Moreover, a non-parametric Friedman test applied to the human preference distributions yields a test statistic of $\chi^2_F = 81.03$ with $3$ degrees of freedom, indicating a high statistical significance ($p \ll 0.001$). This suggests that our sample size is sufficient to reliably detect performance differences.
> > >
> > > # 2. Ego-centric Retrieval Dimensions
> > > Our framework can be flexibly extended to other evaluation dimensions. We have added a "paper quality" dimension using citation counts as the indicator. We constructed **2,640 brand-new queries** (asking models to find the "highest-quality papers" in specific subfields) and evaluated the 8 models. The results demonstrate that Deep Research systems significantly outperform traditional methods when retrieving highly cited papers. We have provided the link to the result chart for this experiment in our previous rebuttal (https://anonymous.4open.science/r/SciNet/Rebuttal/highest-quality.png). We also explicitly show the detailed results in the table below for your immediate convenience:
> > >
> > > | **Category**     | **Models**            | **Quality (Ave. Citations)** |
> > > | ---------------- | --------------------- | ---------------------------- |
> > > | **Embedding**    | SciBERT               | 6.24                         |
> > > |                  | Qwen3-Embed           | 22.34                        |
> > > | **Agentic**      | PaSa                  | 217.93                       |
> > > |                  | PaperQA               | 19.39                        |
> > > |                  | gpt-4o-mini-search    | 4836.04                      |
> > > |                  | gpt-4o-search         | 6272.06                      |
> > > | **DeepResearch** | o4-mini-deep-research | 6678.98                      |
> > > |                  | o3-deep-research      | 7791.59                      |
> > >
> > > # 3. Dataset Generalization and Related Work
> > > In the revised manuscript, we will add a dedicated "Limitations" section. This section will delve into the issue that our constructed queries lack the randomness of real user inputs (wild queries), and will discuss the future direction of expanding diversity by collecting real search logs from academic systems. We also plan to leverage LLMs to rewrite and augment the existing queries, for example, by simulating questioning styles from different professional backgrounds or with varying degrees of colloquialism. These will serve as efficient supplementary approaches to help us build a more challenging test set that closely mirrors real-world application scenarios. Additionally, we will substantially expand Section 7 (Related Work) to provide a detailed comparison between existing retrieval methods and benchmarks, thereby more clearly positioning the unique contributions of SciNet.
> > >
> > > # 4. Intuitive Illustration of the Disruption Index
> > > We have already created an intuitive illustration of the Disruption Index specifically to address this point. This figure visually contrasts incremental research (where subsequent papers cite both the target paper and its references) with disruptive research (where subsequent papers cite _only_ the target paper, initiating a new paradigm) within the citation network structure. You can view this figure via this link (https://anonymous.4open.science/r/SciNet/Rebuttal/Illustration.png). We will include it in the revised manuscript to improve the readability of the paper.
> > >
> > > Thank you once again for your constructive feedback. We hope these detailed results successfully address your remaining concerns. If you have any new insights or further questions, we are more than happy to engage in further in-depth discussions.

---

### Official Review · Reviewer_h1nK · 2026-03-12

**Soundness:** 3
**Presentation:** 3
**Significance:** 3
**Originality:** 3
**Overall Recommendation:** 4
**Confidence:** 4

**Summary:**

This paper introduces SciNet, a benchmark dataset designed to evaluate AI agents for relation-aware scientific literature retrieval. The authors argue that current retrieval systems for scientific literature—such as embedding-based retrieval and agentic research assistants—primarily rely on semantic similarity or keyword matching and therefore fail to capture the relational structure of scientific knowledge.

To address this limitation, the paper proposes SciNet, a dataset built from a large-scale scholarly meta-database containing approximately 269 million scientific papers across seven disciplines. The dataset includes 8,940 tasks that assess a system’s ability to reason about relationships between papers in three categories:
(1) ego-centric tasks, which involve identifying papers with specific scientific properties such as novelty or disruptiveness;
(2) pair-wise tasks, which require identifying relationships between two papers (e.g., whether one supports or contradicts another); and
(3) path-wise tasks, which involve reconstructing citation chains that represent the evolution of scientific ideas.

The authors evaluate eight retrieval systems spanning three paradigms: embedding-based retrieval models, agent-based retrieval pipelines, and deep research systems. Results show that current systems perform poorly on relation-aware tasks, often achieving accuracy below 20% in some settings. The paper further demonstrates that integrating relation-aware retrieval through SciNet improves the quality of generated literature reviews in a downstream application.

The authors release the dataset and evaluation framework to encourage further research on relation-aware scientific retrieval.

**Compliance With Llm Reviewing Policy:**

Affirmed.

**Final Justification:**

The authors provided a thorough and well-organized rebuttal that effectively addressed my main concerns. My assessment remains a Weak Accept.
The paper introduces SciNet, a large-scale relation-aware benchmark for scientific literature retrieval that fills a meaningful gap in existing evaluation frameworks. The task taxonomy (ego-centric, pair-wise, and path-wise) is conceptually well-motivated, and the scale of the underlying corpus (269 million papers) lends credibility to the benchmark's practical relevance. The downstream literature review experiment, showing a ~25% quality improvement, provides concrete evidence of real-world utility.
The rebuttal satisfactorily addressed my concerns about label reliability (98% human-LLM agreement on 200 citation contexts), baseline coverage (spanning established and cutting-edge models across three paradigms), dataset bias (via scale-independent metrics, domain isolation across 2,640 subfields, and cross-temporal pairing), and scalability (millisecond-level latency with SQLite indexing on a 24-core CPU). The commitment to elevating the error analysis to the main text and expanding the limitations and related work sections is appreciated.
My remaining minor concern is that several improvements described in the rebuttal-including expanded error analysis, richer limitations discussion, and the disruption index illustration-are promised revisions rather than changes already reflected in the current manuscript. I encourage the authors to ensure these additions are clearly and fully incorporated in the final version. Overall, I consider this a solid and timely contribution to the field, and I maintain my Weak Accept recommendation.

**Key Questions For Authors:**

Task construction methodology:
How are pair-wise relations such as “support,” “contradict,” or “extend” determined in the dataset? Are these labels automatically generated, manually annotated, or derived from citation contexts?

Dataset validation:
What measures were taken to ensure the accuracy and reliability of task labels? For example, were human evaluations conducted to validate relational annotations?

Baseline coverage:
How were the evaluated retrieval systems selected, and do they represent the strongest available approaches for scientific literature retrieval?

Dataset bias and representativeness:
Given that citation networks can be biased toward highly cited papers and certain disciplines, how does the dataset account for or mitigate these biases?

Scalability considerations:
What are the computational requirements for performing relation-aware retrieval on large scientific corpora, and how feasible is it for real-world research tools?

**Limitations:**

Partially. The paper discusses the difficulty of relation-aware retrieval and limitations of current systems, but it would benefit from a more explicit discussion of dataset biases, potential noise in automated labeling, and limitations of citation-based relational signals.

**Strengths And Weaknesses:**

Strengths

1. Important and timely problem.
The paper addresses a relevant challenge in scientific information retrieval: the inability of current retrieval systems to capture the relational structure of scientific knowledge. As AI agents become more widely used for literature search and automated research assistance, evaluating their ability to reason about scientific relationships is increasingly important.

2. Large-scale dataset construction.
SciNet is constructed using a massive scholarly corpus (OpenAlex), covering hundreds of millions of papers across multiple scientific domains, which gives the dataset strong potential as a benchmark resource for future research.

3. Clear task taxonomy.
The categorization of tasks into ego-centric, pair-wise, and path-wise relational reasoning provides a useful conceptual framework for evaluating literature retrieval systems. This taxonomy helps clarify different levels of relational understanding that retrieval systems might require.

4. Comprehensive evaluation across retrieval paradigms.
The authors evaluate multiple types of systems, including embedding-based retrieval, agentic systems, and deep research pipelines. This comparison helps illustrate the limitations of current approaches in relation-aware retrieval tasks.

5. Downstream application demonstration.
The paper shows that integrating relation-aware retrieval improves literature review quality by a significant margin (reported as ~25%), suggesting practical benefits beyond benchmark performance.

Weaknesses

1. Benchmark-focused contribution with limited methodological novelty.
The primary contribution is the dataset and evaluation framework rather than a new retrieval algorithm. While benchmarks are valuable, the paper may be viewed as incremental unless the dataset demonstrably fills a unique gap not addressed by existing evaluation frameworks.

2. Limited transparency in task construction.
Although the dataset is large, the paper provides limited detail about how tasks are generated and validated. For example, how pair-wise relations such as “support” or “contradict” are determined is not entirely clear, and the reliability of automated labeling could significantly affect benchmark validity.

3. Evaluation metrics and baselines could be stronger.
While several retrieval paradigms are evaluated, it is unclear whether the strongest modern retrieval and reasoning models are included or properly tuned. In addition, the evaluation focuses mainly on accuracy metrics without extensive analysis of error modes or retrieval behavior.

4. Potential dataset biases.
Because the dataset relies heavily on citation networks and bibliometric signals, it may inherit biases such as citation popularity effects, disciplinary differences in citation practices, or temporal skew. These issues are not extensively discussed.

5. Limited analysis of scalability and practical deployment.
Although the dataset is large, the paper does not thoroughly analyze the computational cost of performing relation-aware retrieval at scale, which may limit practical adoption.

---

> ### Author Rebuttal · Authors · 2026-03-31
>
> Thank you for your constructive comments. We address your concerns below:
>
> **Benchmark Contribution (Weakness 1)**
> The core novelty of SciNet lies in introducing relation-aware retrieval tasks from a scientometrics perspective. We do not merely propose a new dataset. Rather, through extensive testing and analysis, we uncover a critical structural flaw in current LLM agents in AI-for-Science contexts: their severe inability to navigate complex scientific networks. By quantifying this blind spot, we provide a clear optimization roadmap for the development of future AI scientists. Therefore, this work is not simply an incremental contribution.
>
>
> **Task Construction & Label Reliability (Weakness 2, Question 1 & 2)**
> To ensure label reliability for pair-wise sentiment relations, we adopt a three-stage pipeline: **rule-based parsing + LLM classification + expert validation**. Specifically, we use GROBID to extract citation contexts from PDFs via in-text citation anchors, and then employ an LLM to classify sentiment polarity.
> To validate the automated labels, we have added an expert evaluation experiment: two AI PhD researchers manually annotated the sentiment of 200 citation contexts. The human–LLM agreement reaches **98%**, with only 4 minor discrepancies on the neutral/positive boundary. This strong agreement indicates that the automated annotations are highly consistent with human judgment.
>
>
> **Baseline Coverage & SOTA Models (Weakness 3, Question 3)**
> To ensure a comprehensive evaluation across the spectrum of modern retrieval paradigms, we selected eight systems across three categories: embedding models, agentic models, and deep research agents. In the embedding category, we included the widely applied classic scientific embedding model SciBERT (5,334 citations) and the recently released, high-capacity Qwen3-Embed (over 40 million downloads on Hugging Face), aiming to cover both well-established and cutting-edge embedding approaches. For agentic pipelines tailored to academic scenarios, we chose PaSa (ACL 2025) and the influential open-source project PaperQA (8.3k GitHub Stars), supplemented by the flagship search model gpt-4o-search, to capture systems capable of autonomous retrieval and multi-step reasoning. Finally, deep research systems (o3-deep-research, o4-mini-deep-research) were included to represent the ceiling of commercial retrieval-augmented reasoning models.
>
>
>
> **Error Analysis (Weakness 3)**
> To explicitly address retrieval failure modes, we will elevate the detailed case analyses from Appendix B to the main text:
> - **Case 1 (DPO Disruption, App. B.1):** Demonstrates models conflating semantics with topology. Embeddings suffered from _Semantic Drift_ (keyword-matching a 1953 paper), while web Agents exhibited _Recency Bias_ (favoring low-impact 2025 preprints), entirely missing true structural milestones like KTO or IPO.
> - **Case 2 (Vision Evolution, App. B.2):** Analyzes long-span reconstruction failures. GPT-4o showed severe _Path Fragmentation_ (leaping over 40 years), and even the SOTA o3-deep-research missed critical citation hubs (e.g., Faster R-CNN, VGGNet) despite generating a semantically coherent narrative.
>
> **Mitigating Dataset Biases (Weakness 4, Question4, Limitations)**
> We have employed a series of rigorous scientometric indicators and careful construction strategies to mitigate the biases potentially introduced by citation networks.
> - **Citation Popularity:** Our core metrics are mathematically scale-independent. The Disruptiveness Index $\frac{N_I - N_J}{N_I + N_J}$ normalizes by total relevant citations, allowing a niche, paradigm-shifting paper to outscore highly-cited incremental work. Similarly, the Novelty metric (Uzzi's Z-score) evaluates backward-looking reference combinations, completely decoupling true innovation from future citation counts.
> - **Disciplinary Differences:** We implemented strict _domain isolation_. By dividing the corpus into 2,640 subfields and evaluating queries independently within them, we eliminate unfair numerical comparisons across disciplines with varying citation baselines.
> - **Temporal Skew:** Our Path-wise tasks utilize cross-temporal pairing (connecting Top-50 Classic papers with Top-10 Emerging post-2024 papers). This forces long-span evolutionary reasoning and neutralizes recency bias.
>
>
> **Scalability and Practical Deployment (Weakness 5, Question5)**
> With proper structured indexing, the computational cost of relation-aware retrieval is surprisingly low. Our actual deployment utilizes a lightweight SQLite index for the 269-million-paper network. Running purely on a 24-core CPU, single-query latency across this massive graph is strictly controlled at the **millisecond level**. For real-world industrial tools, upgrading to enterprise databases (e.g., MySQL, Elasticsearch) or utilizing GPUs would further enhance throughput by orders of magnitude. Thus, large-scale deployment is highly engineering-feasible, cost-effective, and easily scalable.

---

> > ### Author Rebuttal · Reviewer_h1nK · 2026-04-03
> >
> > Thank you for the detailed rebuttal. I appreciate the authors' efforts in addressing my concerns. My concerns are largely resolved, and I maintain my score of Weak Accept.
> > Regarding the key weaknesses I raised: the label reliability concern has been convincingly addressed by the human validation experiment showing 98% human-LLM agreement on 200 citation contexts. The baseline coverage is well-justified, spanning both established and cutting-edge systems across three retrieval paradigms. The response on dataset biases (scale-independent metrics, domain isolation across 2,640 subfields, and cross-temporal pairing) is thorough and satisfactory. The commitment to elevating the error analysis from the appendix to the main text also addresses my concern about the limited analysis of retrieval failure modes. Scalability concerns are resolved by the practical deployment details provided.
> > My remaining minor concern is that several of the improvements described in the rebuttal - such as the expanded error analysis, the limitations discussion, and the enriched related work section - are promised for the revision but not yet reflected in the current manuscript. I encourage the authors to ensure these additions are incorporated clearly in the final version. Overall, I believe this is a solid and timely contribution, and I am comfortable maintaining my Weak Accept recommendation.

---

> > > ### Author Response · Authors · 2026-04-03
> > >
> > > Thank you for your positive feedback and for maintaining the "Weak Accept" score. We are glad that our rebuttal addressed your concerns effectively.
> > >
> > > We have noted your comment regarding the manuscript revision. We confirm that all promised improvements, including the expanded error analysis, limitations discussion, and enriched related work, will be strictly incorporated into the final version.
> > >
> > > Thank you again for your constructive suggestions.

---

### Decision · Program_Chairs · 2026-04-30

**Decision:**

Accept (regular)

**Comment:**

The reviewers are overall positive about this paper. The authors provide a detailed rebuttal to address reviewers’ concerns. Authors are encouraged to incorporate their response in the camera-ready.